

# Performance modelling and scaling of fixed-wing ground-generation airborne wind energy systems

Rishikesh Joshi[1], Roland Schmehl[1], and Michiel Kruijff[2]

[1]Delft University of Technology, Faculty of Aerospace Engineering, Kluyverweg 1, 2629 HS Delft, The Netherlands
[2]Ampyx Power B.V, Lulofsstraat 55, Unit 13, 2521AL Den Haag, The Netherlands

**Correspondence:** Rishikesh Joshi (r.joshi@tudelft.nl)

**Abstract.**

The economic viability of large-scale future airborne wind energy systems critically hinges on the achievable power output in a given wind environment and the system costs. This work presents a fast model for estimating the net power output of fixed-wing ground-generation airborne wind energy systems in the conceptual design phase. In this quasi-steady approach, the kite is represented as a point mass and operated in circular flight manoeuvres while reeling out the tether. This phase is subdivided into several segments. Each segment is assigned a single flight state resulting from an equilibrium of the forces acting on the kite. The model accounts for the effects of flight pattern elevation, gravity, vertical wind shear, hardware limitations, and drivetrain losses. The simulated system is defined by the kite, tether and drivetrain properties, such as the kite wing area, aspect ratio, aerodynamic properties, tether dimensions and material properties, generator rating, maximum allowable drum speed, etc. For defined system and environmental conditions, the cycle power is maximised by optimising the operational parameters for each phase segment. The operational parameters include cycle properties such as the stroke length (reeling distance over the cycle), the flight pattern average elevation angle, and the pattern cone angle, and include segment properties such as the turning radius of the circular manoeuvre, the wing lift coefficient, and the reeling speed. To analyze the scaling behaviour, we present a kite mass estimation model based on the wing area and the maximum tether force. The model mainly aims at sensitivity and scaling studies to support design and innovation trade-offs. It is also suitable for integrating cost models and systems engineering tools that assist in the conceptual design of systems. The computed results are compared with six-degree-of-freedom simulation results of a system with a rated power of $150\,\text{kW}$. The interdependencies between key environmental, system design, and operational parameters are presented. The model's capability to capture scaling effects is shown through an example of varying kite wing area and tether diameter.

## 1 Introduction

Airborne wind energy (AWE) is an emerging technology that uses tethered airborne devices to harness the higher altitude wind resource inaccessible to conventional towered wind turbines with potentially lower material usage. Figure 1 shows several implemented prototypes in the power range of rated powers up to $600\,\text{kW}$. Makani and Kitekraft use onboard ram-air turbines to convert the relative flow at the aircraft into electricity, using a conductive tether to transmit the electricity to the ground.





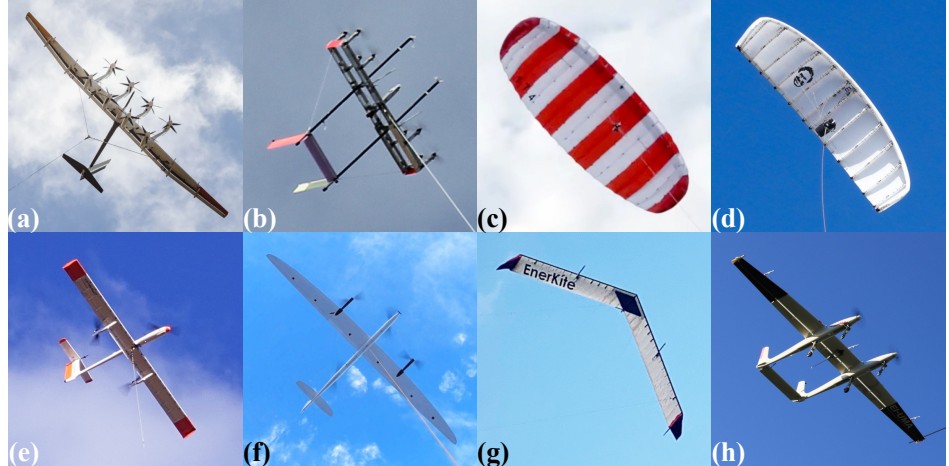

**Figure 1.** Implemented AWE systems: (a) Makani (2020), discontinued in 2020, (b) Kitekraft (2024), (c) SkySails (2024), (d) Kitepower (2024), (e) TwingTec (2024), (f) Kitemill (2024), (g) Enerkíte (2024), and (h) Mozaero (2024), formerly Ampyx Power.

Skysails and Kitepower operate large soft-wing kites in pumping cycles, using suspended cable robots for control and converting the pulling force of the kite into electricity by means of a ground-based drum-generator module. TwingTec and Kitemill use fixed-wing kites that adopt the same conversion principle in combination with the vertical takeoff and landing (VTOL) subsystem. Enerkíte operates a lightweight fixed-wing kite in pumping cycles, using three tethers controlled from the ground station. A rotational mast on the ground station is employed to launch the kite. Mozaero, formerly operating under the name

Ampyx Power, uses a catapult subsystem combined with onboard propellers to launch the kite.

This work focuses on the fixed-wing ground-generation (ground-gen) concept developed by Ampyx Power, which has been continued by Mozaero since 2023. As illustrated in Fig. 2, a fixed-wing kite analogous to a glider aircraft is connected by a tether to a drum-generator module on the ground. The kite flies in repetitive crosswind patterns, pulling the tether with high force from the drum and driving the generator, as shown in Fig. 2(a). During this reel-out phase, electricity is generated.

Once the tether has reached a certain length, the kite is retracted towards the generator with minimum aerodynamic drag and substantially lower force as shown in Fig. 2(b). A small fraction of the generated electricity is consumed during this reel-in phase. An intermediate buffer storage is typically used for this purpose. The reel-out and reel-in phases are repeated cyclically to generate a net power output.

Compared to flexible membrane kites, fixed-wing kites are characterized by better aerodynamic performance, a higher lift-

to-drag ratio, and a substantially larger mass-to-wing surface ratio. While the first ratio stays roughly the same when increasing the size of the kite, the latter ratio progressively increases. This increase affects fixed-wing kites much more than it does soft-wing kites, rendering mass a crucial parameter for designing large-scale fixed-wing AWES. The present work focuses on the gravitational effect of mass and its interplay with the resultant aerodynamic force and the tether force during quasi-steady flight operation. To understand this effect, it is important to note that the gravitational force is a constant contribution to the force



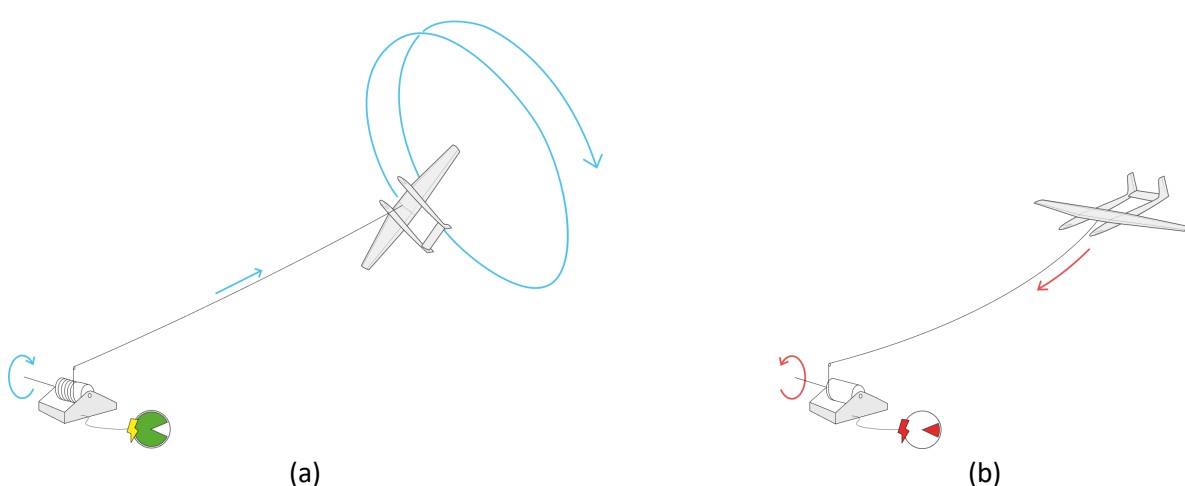

**Figure 2.** Operation schematic of the fixed-wing ground-generation airborne wind energy concept: (a) Reel-out phase, (b) Reel-in phase (image courtesy of Ampyx Power B.V.).

equilibrium at the kite. In contrast, the aerodynamic force depends on the instantaneous flight speed and can thus vary greatly in the different operational phases of the system. The tether force, on the other hand, is a reaction force to the vectorial sum of the two external forces.

Because of the fast crosswind manoeuvres in the reel-out phase, the apparent wind speed is high, and the quasi-steady force equilibrium is dominated by the aerodynamic loading and the tether force - except for low wind speeds at cut-in where the gravitational force contributes substantially. In the reel-in phase, the gravitational effect is exploited to retract the kite with a minimum tether force and thus reel-in power. The force equilibrium is dominated by the aerodynamic and gravitational forces, while the tether force plays only a minor role. Several different strategies exist for launching and landing. For AWE systems pursuing a vertical takeoff and landing (VTOL) strategy, the aerodynamic force generated by the VTOL subsystem is used entirely to overcome gravity because the wing is either perpendicular to the wind and thus ineffective for lift (e.g. Makani or Kitekraft) or aligned with the wind but providing an insufficient lift force (e.g. Kitemill or TwingTec). For AWE systems pursuing a horizontal takeoff and landing (HTOL) strategy, the kite needs to be accelerated by an external mechanism to a certain minimum flight speed at which the aerodynamic force can overcome gravity. This can be done with a catapult and optional onboard propellers (e.g. Ampyx Power) or a swivelling mast (Enerkíte). Irrespective of how the fixed-wing kite is launched or landed, it has to maintain a certain minimum flight speed during crosswind operation to create an aerodynamic load level sufficient to compensate for gravity's effect and stay airborne. This interplay between aerodynamic, gravitational, and tether forces during the different operational phases requires careful tradeoff analysis when designing a fixed-wing AWE system.

Several physical models with a broad spectrum of fidelity and scope have been developed to understand and mathematically describe the operation of AWE systems. Higher-fidelity approaches based on dynamic models and system control, such as





in Licitra et al. (2019); Malz et al. (2019); Eijkelhof and Schmehl (2022) are computationally expensive and require the initialisation and tuning of many parameters. Sommerfeld et al. (2022) investigated the scaling effects of fixed-wing ground-generation AWE systems using AWEbox (Schutter et al., 2023), which is an optimal control framework. Their simulation results do not reveal consistent trends which could indicate non-converged results. The results of such models are highly dependent on the implemented controller's performance. Hence, they are not the best option for understanding the fundamental

principles of systems, the achievable energy output, and the interdependencies between environmental, system design, and operational parameters. Lower-fidelity approaches based on steady or quasi-steady models can be used for this purpose, and techno-economic analysis, such as in Heilmann and Houle (2013); Faggiani and Schmehl (2018); Joshi et al. (2023). There have been several attempts to model the power generation characteristics of AWE systems with lower fidelity, but none of the theories account for all the relevant physical effects. The first mathematical foundation for estimating power extraction

using tethered kites in a crosswind motion was laid by Loyd (1980). This analytical theory assumes idealised flight states to estimate the mechanical power output of a pumping AWE system, but it does not account for losses due to elevation, retraction phase, gravity, vertical wind shear, and hardware limitations. Argatov et al. (2009) extended this crosswind theory to spherical coordinates to compute the mean mechanical reel-out power of ground generation systems. The theory accounts for the averaged effects of elevation and gravity on the kite but does not account for the losses due to the retraction phase.

Luchsinger (2013) extended Loyd's ideal power extraction theory for fixed-wing ground generation systems to account for the retraction phase, average pattern elevation and hardware limitations such as the maximum tether force and generator power rating. The study did not account for the effect of gravity. Fechner and Schmehl (2013) presented a model for soft-wing ground generation systems that accounts for losses due to elevation, the transition phase, and the retraction phase but did not account for the effect of gravity. They also accounted for various efficiencies of the ground station in computing net electrical power

output. The model results were compared against measurements of a $4\,\mathrm{kW}$ prototype demonstrator. Ranneberg et al. (2018) presented a model to compute the net power output that accounts for the traction and retraction phases together but does not account for the effects of gravity. The model results were compared against measurements of a $5\,\mathrm{kW}$ prototype, which showed deviations within their expected range. Trevisi et al. (2020) presented an analytical modelling framework to compute net power output, accounting for the losses due to elevation, gravity, and the retraction phase. Following this work, Trevisi

et al. (2023) proposed refining the power estimation by including the effect of far wake on the induced drag. These losses were modelled as loss factors to calculate the net power output. The model results were not compared against higher fidelity simulations or measurements. A steady-state equilibrium model for soft-wing ground generation systems was developed by Schmehl et al. (2013); Van der Vlugt et al. (2019) and validated by Schelbergen and Schmehl (2020) using measurements from a $20\,\mathrm{kW}$ prototype. This model accounts for the losses due to elevation, retraction phase, transition phase and gravity.

We have extended this formulation to fixed-wing systems by incorporating changes in the retraction phase, angle of attack controllability impacting the operational lift coefficient, consideration of induced drag, effects of vertical wind shear, hardware limitations, and drivetrain losses. The model is then used to formulate an optimisation problem to find the operational set points for maximising the system's electrical cycle power.





The paper is structured as follows. Section 2 describes the theoretical framework, which begins with simplifying assumptions
and then builds up by relaxing them step by step, Sect. 3 presents the model capabilities through a numerical example, and
Sect. 4 presents the conclusions.

## 2 Model Description

Consider a fixed-wing kite, analogous to a glider airplane, with wing planform area $S$, aspect ratio $\mathit{R}$, lift and drag coefficients
$C_L$ and $C_D$, respectively, with a tether of maximum allowable force $F_{t,max}$, flying on a circular flight trajectory, with a turning
radius of $R_p$, cone opening angle $\gamma_p$, and turning axis elevation angle $\beta_p$, which is also referred to as the pattern elevation
angle in the following sections. The pattern elevation angle is kept constant for one reel-out and reel-in phase. This is the
first step of a system-level performance analysis, so unsteady effects such as turbulence or wind gusts are not considered. The
kite kinematics and the forces acting on the kite are formulated in a spherical reference frame $(r, \theta, \phi)$, defined with respect
to the Cartesian wind reference frame $(X_w, Y_w, Z_w)$. The horizontal axis is aligned with the wind velocity $\mathbf{v}_w$ and its Z-axis
pointing vertically upwards, as shown in Fig. 3. The unit vectors $\mathbf{e}_r, \mathbf{e}_\theta$, and $\mathbf{e}_\phi$ define a right-handed local vector base. The
kite's position is represented by point $\mathbf{K}$, and the ground station is located at the origin $\mathbf{O}$. The radial coordinate $r$ specifies
the geometrical distance between the kite and the ground station, $\theta$ is the polar angle which is complementary to the tether
elevation angle $\beta$ measured from the ground (i.e. $\theta + \beta = 90°$), and $\phi$ is the azimuth angle.

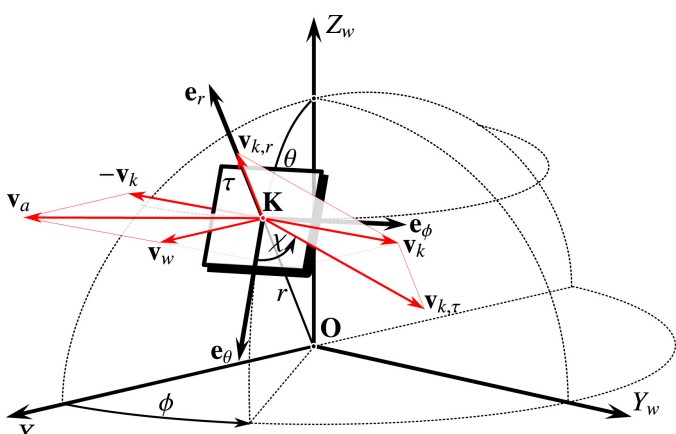

**Figure 3.** Decomposition of kite kinematics in a spherical reference frame (Schmehl et al., 2013).

The kite velocity $\mathbf{v}_k$ can be decomposed into radial and tangential components $v_{k,r}$ and $v_{k,\tau}$, respectively. The direction of
$v_{k,\tau}$ is given by the course angle $\chi$ measured in the local tangential plane $\tau$ from the unit vector $\mathbf{e}_\theta$. The apparent wind velocity



can be expressed in spherical coordinates $(r, \theta, \phi)$ as

$$\mathbf{v}_a = \begin{bmatrix} v_{a,r} \\ v_{a,\theta} \\ v_{a,\phi} \end{bmatrix} = \begin{bmatrix} \sin\theta\cos\phi \\ \cos\theta\cos\phi \\ -\sin\phi \end{bmatrix} v_w - \begin{bmatrix} 1 \\ 0 \\ 0 \end{bmatrix} v_{k,r} - \begin{bmatrix} 0 \\ \cos\chi \\ \sin\chi \end{bmatrix} v_{k,\tau}. \tag{1}$$

The final model simulating the reel-out and reel-in phases of a system is formulated as an optimisation problem where the net electrical cycle power is maximised for given wind conditions. The fixed model inputs are the system design parameters, such as the kite wing area, aspect ratio, wing aerodynamic properties, tether properties, speed limits of the drum-generator module, etc. The optimisation variables are the operational parameters such as the stroke length (reeling distance over a cycle), pattern elevation angle, cone opening angle, turning radius at the start of the cycle, kite reeling speed and operating lift coefficient. The optimisation problem is constrained by physical limits such as the maximum tether force, tether length limit, minimum ground clearance, maximum operation height, etc.

Section 2.1 describes power extraction using the assumption of a massless kite on a straight tether at an elevation. Section 2.2 introduces a kite mass estimation function based on prototype data and scaling laws. Section 2.3 incorporates the effect of gravity on the power output. Section 2.4 describes the impact of the retraction phase on the net power output of the system. Section 2.5 introduces vertical wind shear, and finally, Section 2.6 describes the electrical cycle power of a system considering all the effects together.

## 2.1 Massless kite at an elevation

Figure 4(a) illustrates the physical problem of a massless kite with a straight inelastic tether flying a circular pattern. In the depicted situation, the kite just passes through the $\mathbf{X}_w\mathbf{Z}_w$-plane and the wind vector $\mathbf{v}_w$ is orthogonal to the kite's tangential motion component. At this analysis stage, we assume a uniform and constant wind field, i.e. the wind speed and the direction do not change in time and space. The rotation of the wind reference frame is assumed to be so slow that the accelerations induced by this rotation are negligible. For any arbitrary point on the circular flight manoeuvre, Fig. 4(b) shows the decomposition of velocity and force vectors in the $\mathbf{e}_r\mathbf{v}_a$-plane.

The assumption of a massless kite allows us to ignore the effects of gravity and inertia on the kite's motion. Since the tether cannot support a bending moment, the radial force balance is

$$F_t = F_a, \quad \text{where} \tag{2}$$

$$F_a = \frac{1}{2}\rho S \sqrt{C_L^2 + C_D^2}\, v_a^2. \tag{3}$$

Because of the assumed straight tether, the kite's radial speed $v_{k,r}$ is identical to the reel-out speed $v_o$. For any point on the flight trajectory, the apparent wind speed can be expressed by its radial and tangential components as

$$v_a = v_{a,r}\sqrt{\left(\frac{v_{a,\tau}}{v_{a,r}}\right)^2 + 1}. \tag{4}$$



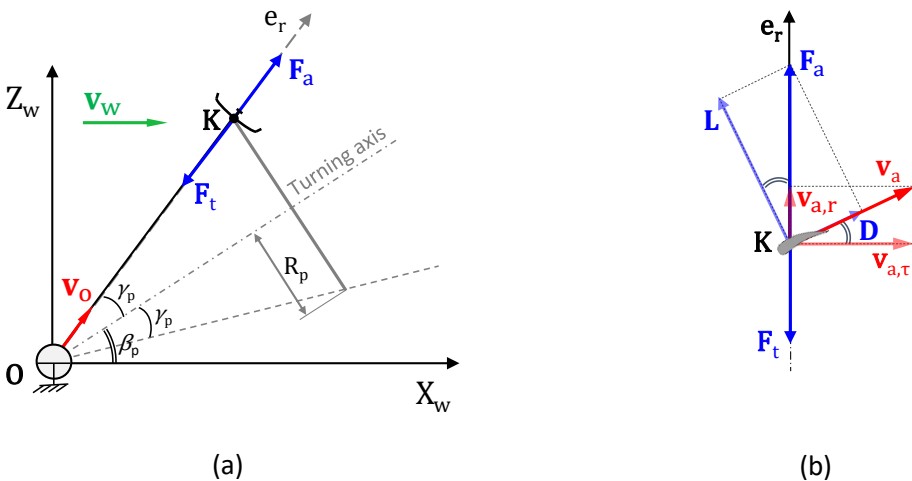

**Figure 4.** Velocities and forces for the massless kite at an elevation. (a) Side view illustrating the circular flight manoeuvre with average pattern elevation angle $\beta_\mathrm{p}$, opening cone angle $\gamma_\mathrm{p}$, reel-out speed $v_\mathrm{o}$, and force equilibrium $\mathbf{F}_\mathrm{t} + \mathbf{F}_\mathrm{a} = 0$ at the kite. (b) Decomposition of velocity and force vectors in the $\mathbf{e}_\mathrm{r}\mathbf{v}_\mathrm{a}$-plane of the spherical reference frame for any arbitrary point on the trajectory.

The radial component is defined by Eq. (1) as

$$v_{\mathrm{a,r}} = v_\mathrm{w} \sin\theta \cos\phi - v_\mathrm{o}. \tag{5}$$

The ratio of tangential and radial components of the apparent wind velocity is also known as the kinematic ratio as described in Schmehl et al. (2013). Because of the geometric similarity of velocity and force triangles illustrated in Fig. 4(b), the kinematic ratio can be related to the lift-to-drag ratio,

$$\kappa = \frac{v_{\mathrm{a,\tau}}}{v_{\mathrm{a,r}}} = \frac{C_\mathrm{L}}{C_\mathrm{D}}. \tag{6}$$

From Eqs. (2) to (6), the extractable mechanical power at the ground station can be computed as

$$P_{\mathrm{m,o}} = F_\mathrm{t} v_\mathrm{o} \tag{7}$$

$$= \frac{1}{2} \rho S \sqrt{C_\mathrm{L}^2 + C_\mathrm{D}^2} (v_\mathrm{w} \sin\theta \cos\phi - v_\mathrm{o})^2 \left[ \left( \frac{C_\mathrm{L}}{C_\mathrm{D}} \right)^2 + 1 \right] v_\mathrm{o}. \tag{8}$$

From this equation, one can conclude that increasing values of the elevation angle and azimuth angles decrease the magnitude of the apparent wind velocity vector. We simplify our formulation and represent one circular flight manoeuvre by a single flight state. During one full circular flight manoeuvre, the elevation angle $\beta$ will vary from $\beta_\mathrm{p} - \gamma_\mathrm{p}$ to $\beta_\mathrm{p} + \gamma_\mathrm{p}$, where $\gamma_\mathrm{p}$ is the cone opening angle. The geometric average $\beta_\mathrm{p}$ over one full manoeuvre can be considered a representative elevation angle over the pattern. The effect of the azimuth angle differs from that of the elevation angle. Using the geometric average of the variation of





the azimuth angle as a representative angle will result in $\phi_\mathrm{p} = 0$ due to the negative and positive signs of the azimuth angle on either side of the symmetry plane $X_\mathrm{w} Z_\mathrm{w}$. But in reality, the kite flying on a circular manoeuvre is, on average, at a non-zero azimuth angle (Van der Vlugt et al., 2019). Therefore, we consider the geometric centre of the semicircle as a representative azimuth angle. The centroid $y_\mathrm{c}$ of a semicircle with radius $R$ in the Cartesian reference frame is given by

$$y_\mathrm{c} = \frac{4R}{3\pi}. \tag{9}$$

This translates to a specific azimuth angle in the spherical reference frame. For a given cone angle $\gamma_\mathrm{p}$, the azimuth angle representing the centroid of a semicircle is

$$\phi_\mathrm{p} = \sin^{-1}\left( \frac{4\sin\gamma_\mathrm{p}}{3\pi} \right) \tag{10}$$

Therefore, it can be approximated that a representative point for the entire pattern, incorporating the average effect of elevation and azimuth is with $\theta = \pi/2 - \beta_\mathrm{p}$ and $\phi_\mathrm{p} = \sin^{-1}(4\sin\gamma_\mathrm{p}/3\pi)$. The mean pattern reel-out power can now be estimated using Eq. (7). The reel-out speed is an independent variable in our model, which is controlled by the ground station, and the tangential velocity is a result of the local force balance at the kite. The other dependent properties of the system are the tether dimensions, the kite's operational envelope, and the effective drag coefficient. They are computed as follows.

**Tether dimensions**

For a given turning radius $R_\mathrm{p}$ and opening cone angle $\gamma_\mathrm{p}$ during operation, the required tether length is computed as

$$l_\mathrm{t} = \frac{R_\mathrm{p}}{\sin\gamma_\mathrm{p}}. \tag{11}$$

For a given tether tensile strength $\sigma_\mathrm{t}$ and maximum allowable tether force $F_\mathrm{t,max}$, the required tether diameter can be calculated as

$$d_\mathrm{t} = \sqrt{\frac{4F_\mathrm{t,max}}{\pi\sigma_\mathrm{t}}}. \tag{12}$$

For ground-gen systems, the tether lifetime is driven primarily by fatigue due to bending and creep (Bosman et al., 2013). Hence, the tether will not be sized based on the material's ultimate breaking strength but on the optimal operating stress levels for an extended fatigue life.

**Kite's operational envelope**

From Eq. (7), one can see that the closer the tether is aligned with the wind velocity, the higher the power generated in crosswind operation. However, in a practical operation scenario, the kite must maintain a certain ground clearance $h_\mathrm{min}$, and in most cases, respect a maximum operating height $h_\mathrm{max}$ for safety reasons and regulations. These limits must be considered when estimating the tether length and the operational height during the cycle. Figure 5 shows this geometrical relationship where $z_\mathrm{k,min}$ and $z_\mathrm{k,max}$ are the bottom-most and the top-most operating points during the cycle, $l_\mathrm{t,min}$ and $R_\mathrm{p,min}$ are the tether length and the



turning radius at the start of the cycle, respectively. The operational height range of the kite is the unhashed vertical region between $h_{\min}$ and $h_{\max}$. The region between the red dotted lines represents the actual operational envelope of the kite. This envelope changes since the optimal operating parameters change with respect to different wind conditions.

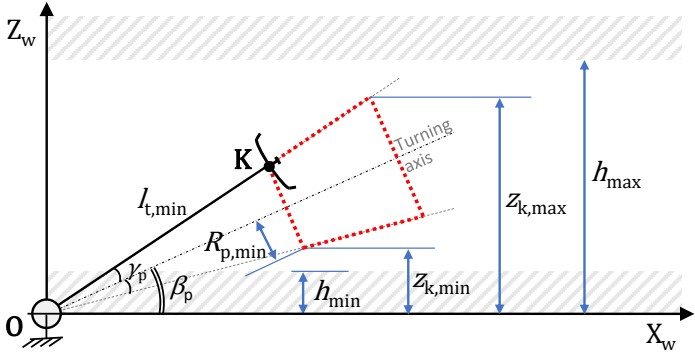

**Figure 5.** Side view illustrating the kite's operational envelope (region shown by the red dotted lines), and the operational height range (unhashed region between $h_{\min}$ and $h_{\max}$).

Determining the minimum tether length $l_{\mathrm{t,min}}$ using Eq. (11), the minimum ground clearance $h_{\min}$ can be enforced by
190 computing the kite's height at the bottom-most point of its circular flight manoeuvre

$$z_{\mathrm{k,min}} = \frac{R_{\mathrm{p,min}}}{\sin \gamma_{\mathrm{p}}} \sin (\beta_{\mathrm{p}} - \gamma_{\mathrm{p}}), \tag{13}$$

such that $z_{\mathrm{k,min}} \geq h_{\min}$. The maximum operational height limit $h_{\max}$ can be enforced similarly by computing

$$z_{\mathrm{k,max}} = \frac{R_{\mathrm{p,max}}}{\sin \gamma_{\mathrm{p}}} \sin (\beta_{\mathrm{p}} + \gamma_{\mathrm{p}}). \tag{14}$$

**Wing lift coefficient**

195 We assume complete control over the kite by modulating the angle of attack to maintain the necessary wing lift coefficient $C_{\mathrm{L}}$ for optimal flight. Therefore, $C_{\mathrm{L}}$ is a variable in our model whose value is based on the lift polar. The maximum lift coefficient will set the upper limit considering some stall margins. This can be obtained from experimental measurements or computational analysis such as in Vimalakanthan et al. (2018).

**Effective drag coefficient**

200 In addition to the wing drag, the tether is responsible for drag losses during operation. The tether drag contribution is assumed to be lumped to the kite, resulting in an effective drag of the combined kite and tether system (Houska and Diehl, 2006). To estimate this effective drag, it is assumed that the generated moment at the ground station equals the sum of the moments generated by the kite and the tether drag individually. The generated drag force is approximately perpendicular to the tether for high lift-to-drag ratios. We assume that the apparent velocity of the topmost segment of the tether is the same as that of the kite,





and it uniformly drops to zero at the ground station. For a given tether length $l_\mathrm{t}$, this moment equality can be mathematically expressed as

$$l_\mathrm{t} D_\mathrm{eff} = l_\mathrm{t} D_\mathrm{k} + \int_0^{l_\mathrm{t}} l \frac{1}{2} \rho v_{\mathrm{a},l}^2 C_\mathrm{d,t} d_\mathrm{t} \mathrm{d}l \tag{15}$$

$$= l_\mathrm{t} \frac{1}{2} \rho v_\mathrm{a}^2 C_\mathrm{D,k} S + \frac{1}{2} \rho \frac{v_\mathrm{a}^2}{l_\mathrm{t}^2} C_\mathrm{d,t} d_\mathrm{t} \int_0^{l_\mathrm{t}} l^3 \mathrm{d}l, \tag{16}$$

where $C_\mathrm{D,k}$ is the kite drag coefficient, $v_{\mathrm{a},l}$ is the apparent velocity of the tether element $\mathrm{d}l$ at a distance $l$ from the ground station, $d_\mathrm{t}$ is the cross-sectional diameter, and $C_\mathrm{d,t}$ is the cross-sectional drag coefficient of the tether. This equation can be solved to estimate the effective drag coefficient

$$C_\mathrm{D,eff} = C_\mathrm{D,k} + C_\mathrm{D,t}, \tag{17}$$

where,

$$C_\mathrm{D,t} = \frac{1}{4} C_\mathrm{d,t} d_\mathrm{t} l_\mathrm{t} \frac{1}{S}. \tag{18}$$

It is the effective drag coefficient of the tether lumped at the kite. The total drag of a wing is the sum of parasitic drag and lift-induced drag. Parasitic drag is comprised of a pressure drag contribution due to flow separation and a skin friction drag contribution. The induced drag is coupled to the generated lift (Anderson, 2016). For a given wing with aspect ratio $Æ\!R$ and wing planform efficiency (Oswald) factor $e$, the total kite drag coefficient can be expressed as

$$C_\mathrm{D,k} = C_\mathrm{d,min} + \frac{(C_\mathrm{L} - C_\mathrm{l,Cd,min})^2}{\pi Æ\!R e}, \tag{19}$$

where $C_\mathrm{d,min}$ is the parasitic drag, $C_\mathrm{L}$ is the wing lift coefficient, and $C_\mathrm{l,Cd,min}$ is the lift coefficient at $C_\mathrm{d,min}$. As stated earlier, the drag polar can be obtained from experimental measurements or computational analysis such as in Vimalakanthan et al. (2018).

### 2.2 Effective mass estimate

Equation (2) does not consider the effect of gravity in the force equilibrium. This effect can be generally neglected during the reel-out phase for smaller systems, especially for low-mass soft-wing systems. This is because the gravitational force is much lower than the traction force. The main impact of weight for soft-wing systems is during the reel-in phase since they typically fly to higher heights because the lift-to-drag ratio is limited to a lower value. Gravity helps to reduce this height and shorten the reel-in phase (Van der Vlugt et al., 2019). This effect differs for larger and fixed-wing systems with higher mass (Eijkelhof and Schmehl, 2022). Gravity reduces the attainable tether force and should be accounted for in the power extraction.

Kruijff and Ruiterkamp (2018); Bonnin (2019) developed a model for mass scaling at the part level based on the $150\,\mathrm{kW}$ prototype AP3 and the MW level concept study AP4 developed by Ampyx Power B.V. The model uses the prototype as a reference system and applies known scaling laws for each structural part within the kite. The reference prototype was designed to





meet aviation standards with relatively conservative safety factors. The prototype's architecture is scalable using a conventional design with ribs, spar caps, webs, etc. It is, therefore, assumed to be a good representation even for much larger fixed-wing kites. In the resulting mass model, the kite mass $m_\mathrm{k}$ is a non-linear function of the kite planform wing area $S$, aspect ratio $Æ\!R$, and maximum tether force $F_\mathrm{t,max}$, given as

$$m_\mathrm{k} = \left[\left(0.024\frac{F_\mathrm{t,max}}{S} + 0.1\right)S^2 + \left(1.7\frac{F_\mathrm{t,max}}{S} + 32.5\right)S - 50\right]\left[0.46\left(\frac{Æ\!R}{Æ\!R_\mathrm{ref}}\right)^2 - 0.66\left(\frac{Æ\!R}{Æ\!R_\mathrm{ref}}\right) + 1.2\right], \quad (20)$$

where $Æ\!R_\mathrm{ref} = 12$, $F_\mathrm{t,max}$ is in kN, $S$ is in m$^2$, and $m_\mathrm{k}$ is in kg. The physical meaning of the term $F_\mathrm{t,max}/S$ corresponds to the wing loading for an aircraft. It states the maximum force it can handle per unit wing area. It varies around $1\text{-}10\,\mathrm{kN\,m^{-2}}$ based on the purpose and size of the aircraft. Figure 6 is a plot of kite mass against wing area using the above equation and fixing $Æ\!R = 12$ and $F_\mathrm{t,max}/S = 3.5\,\mathrm{kN\,m^{-2}}$. These fixed values represent the AP3 $150\,\mathrm{kW}$ prototype values. The result is compared against various data points compiled in Joshi et al. (2023). As seen from the aircraft scaling curve (Roskam, 1989), a conventional aircraft wing does not increase in mass as drastically as an AWE system kite. Based on available information, the MegAWES kite has $Æ\!R = 12$, and $F_\mathrm{t,max}/S = 11\,\mathrm{kN\,m^{-2}}$ (Eijkelhof and Schmehl, 2022); AP2 has $Æ\!R = 10$; AP4 has $Æ\!R = 12$ and $F_\mathrm{t,max}/S = 3\,\mathrm{kN\,m^{-2}}$ (Kruijff and Ruiterkamp, 2018; Ruiterkamp and Sieberling, 2013); AP5 has $Æ\!R = 9.6$ and $F_\mathrm{t,max}/S = 3.9\,\mathrm{kN\,m^{-2}}$ (Hagen et al., 2023); M600 has $Æ\!R = 20$ and $F_\mathrm{t,max}/S = 7.3\,\mathrm{kN\,m^{-2}}$; MX2 has $Æ\!R = 12.5$ and $F_\mathrm{t,max}/S = 4.6\,\mathrm{kN\,m^{-2}}$ (Echeverri et al., 2020); Haas et al. (2019) has $Æ\!R = 26$ and $F_\mathrm{t,max}/S = 12.4\,\mathrm{kN\,m^{-2}}$.

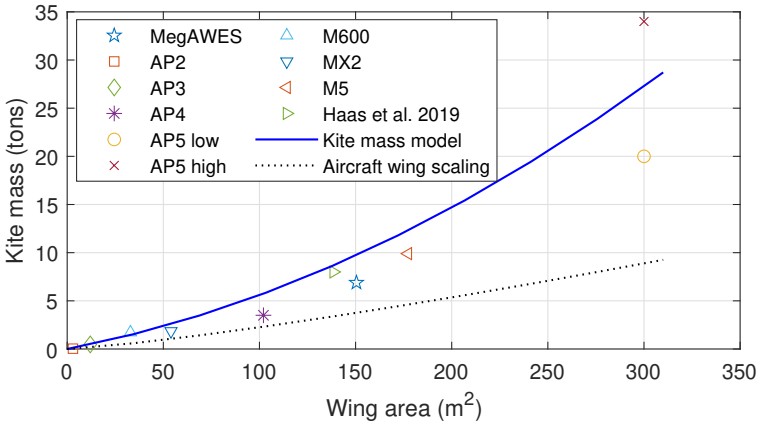

**Figure 6.** Kite mass as a function of wing area. Discrete data points from Joshi et al. (2023). MegAWES: Eijkelhof and Schmehl (2022), Ampyx Power AP2, AP3, and AP4: Kruijff and Ruiterkamp (2018); Ruiterkamp and Sieberling (2013), AP5 low and AP5 high: Hagen et al. (2023), Makani Power M600, MX2 and M5: Hardham (2012); Echeverri et al. (2020), Haas et al. 2019: Haas et al. (2019), Aircraft wing scaling: Roskam (1989).

Figure 7 shows a 3D scatter plot by varying all three parameters in Eq. (20). As expected, the kite mass increases with increasing values of wing area, maximum allowable tether force and aspect ratio.

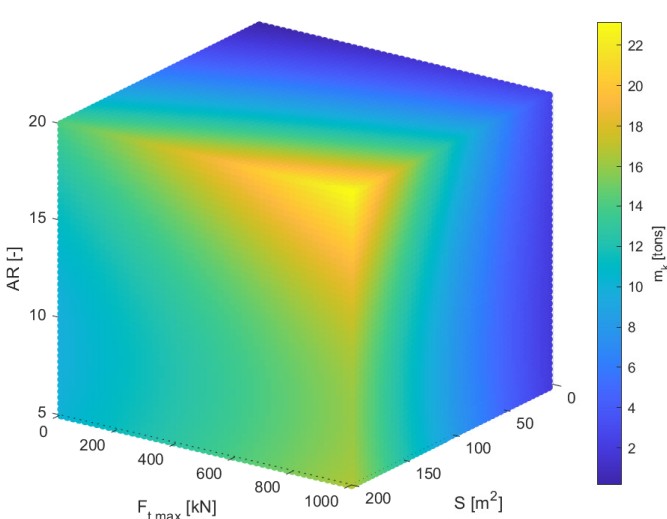

**Figure 7.** 3D scatter plot illustrating the relationship between wing area $S$, maximum tether force $F_{\text{t,max}}$, and aspect ratio $Æ\!R$, with the kite mass $m_{\text{k}}$.

The minimum wind speed required by a static kite to compensate for the gravitational force and stay airborne, also known as the static take-off limit (STOL), will help us further understand the effect of mass. The STOL can be calculated as

$$v_{\text{w,STOL}} = \sqrt{\frac{2m_{\text{k}}g}{\rho S C_{\text{L,max}}}}, \tag{21}$$

where $C_{\text{L,max}}$ is the maximum operable wing lift coefficient. Table 1 shows the STOL for some of the prototypes. When the
wind speeds are below the STOL for the particular kite, for example, during take-off, they need to be accelerated to achieve a higher apparent wind speed that allows compensation for the gravitational force. This implies that less heavy soft-wing kites, such as the TU Delft V3, could be launched at lower wind speeds without additional accelerating mechanisms for take-off. On the other hand, the heavier fixed-wing kites will always need to be accelerated to increase their apparent speeds.

In addition to the kite mass, tether mass also affects power extraction and is assumed to be lumped together with the kite
mass. To determine the equivalent lumped mass, we require that the moment generated at the drum equal the sum of the moments generated by the kite and the continuous tether individually. For a specific tether length $l_{\text{t}}$ and elevation angle $\beta$

$$l_{\text{t}} m_{\text{eff}} \text{g} \cos\beta = l_{\text{t}} m_{\text{k}} \text{g} \cos\beta + \int\limits_{0}^{l_{\text{t}}} l \text{g} \cos\beta \frac{\pi}{4} d_{\text{t}}^2 \rho_{\text{t}} \, \text{d}l, \tag{22}$$

where g is the gravitational acceleration, $d_{\text{t}}$ is the tether diameter, $\rho_{\text{t}}$ is the tether material density, and $\text{d}l$ is the length of the tether element at a distance $l$ from the drum. From this equation, the effective mass can be calculated as

$$m_{\text{eff}} = m_{\text{k}} + \frac{1}{2} m_{\text{t}}, \tag{23}$$





**Table 1.** Static take-off limits for different prototypes. AP2: Kruijff and Ruiterkamp (2018); Williams et al. (2019), AP3: Vimalakanthan et al. (2018); Kruijff and Ruiterkamp (2018), MegAWES: (Eijkelhof and Schmehl, 2022), MX2: Hardham (2012), TU Delft V3: Oehler and Schmehl (2019).

| Kite | Wing area $(\mathrm{m}^2)$ | Kite mass (kg) | $C_{\mathrm{L,max}}(-)$ | STOL $(\mathrm{m\,s}^{-1})$ |
|------|-----------|-----------|-----------|------|
| AP2 | 3 | 35 | 1.5 | 11.3 |
| AP3 | 12 | 475 | 2.1 | 17.5 |
| MegAWES | 150.45 | 6885 | 1.9 | 19.8 |
| MX2 | 54 | 1850 | 2 | 16.7 |
| TU Delft V3 | 19.75 | 22.8 | 1 | 4.3 |

where the mass $m_{\mathrm{t}}$ varies with the deployed length of the tether during the cycle.

## 2.3 Effect of gravity

If we consider the top point of the pattern during operation, shown in Fig. 8, the aerodynamic force vector $\mathbf{F}_{\mathrm{a}}$ has to tilt upwards to compensate for the kite's weight $\mathbf{F}_{\mathrm{g}}$. This tilt is achieved by rolling the kite by an angle $\Psi$ from the radial direction.

In this model, the roll and the pitch are defined as orientation properties of the aerodynamic force vector relative to the radial direction. Along the manoeuvre, the aerodynamic force vector will continuously roll and pitch to counteract gravity. This effectively reduces the contribution of $\mathbf{F}_{\mathrm{a}}$ to $\mathbf{F}_{\mathrm{t}}$. Since $\mathbf{F}_{\mathrm{g}}$ always points downwards, it does not have a component in the $\mathbf{e}_{\phi}$ direction. For the top point of the pattern, the quasi-steady force balance of $\mathbf{F}_{\mathrm{t}}$, $\mathbf{F}_{\mathrm{g}}$ and $\mathbf{F}_{\mathrm{a}}$ in spherical coordinates is

$$
\begin{bmatrix} -F_{\mathrm{t}} \\ 0 \\ 0 \end{bmatrix} + \begin{bmatrix} -F_{\mathrm{g}}\cos\theta \\ F_{\mathrm{g}}\sin\theta \\ 0 \end{bmatrix} + \begin{bmatrix} F_{\mathrm{a,r}} \\ F_{\mathrm{a},\theta} \\ 0 \end{bmatrix} = 0. \tag{24}
$$

Due to the tilting of $\mathbf{F}_{\mathrm{a}}$ relative to the radial direction, the geometric similarity between the velocity and force triangles, as used to formulate Eq. (6), is lost. Hence, the kinematic ratio $\kappa$ cannot be substituted with the glide ratio. The mechanical reel-out power, in this case, becomes

$$
P_{\mathrm{m,o}} = F_{\mathrm{t}} v_{\mathrm{o}} \tag{25}
$$

$$
= \left( \sqrt{\left(\frac{1}{2}\rho S\right)^2 (C_{\mathrm{L}}^2 + C_{\mathrm{D}}^2)(v_{\mathrm{w}}\sin\theta\cos\phi - v_{\mathrm{o}})^4 (\kappa^2 + 1)^2 - (F_{\mathrm{g}}\sin\theta)^2} - F_{\mathrm{g}}\cos\theta \right) v_{\mathrm{o}}. \tag{26}
$$

As explained in Schmehl et al. (2013) and Van der Vlugt et al. (2019), while maintaining the force balance given in Eq. (24), there should be a solution for the kinematic ratio $\kappa$ for which the decomposition of $\mathbf{F}_{\mathrm{a}}$ in $\mathbf{L}$ and $\mathbf{D}$ components corresponds to



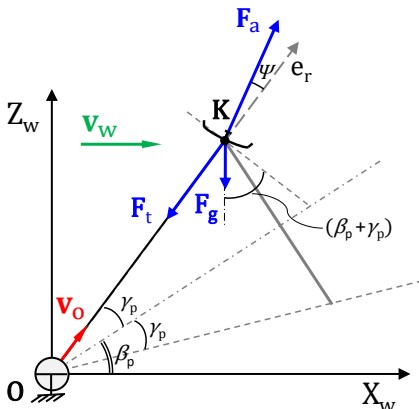

**Figure 8.** Side view illustrating the forces and velocities, including weight, during the reel-out phase of the kite at the top point of its circular manoeuvre.

the glide ratio. By the definition of drag force, $D = (\mathbf{F}_a \cdot \mathbf{v}_a)/v_a$, and using $F_a = \sqrt{L^2 + D^2}$, we get the equation

$$\frac{L}{D} = \sqrt{\left(\frac{F_a v_a}{\mathbf{F}_a \cdot \mathbf{v}_a}\right)^2 - 1}. \tag{27}$$

This equation is a consistency constraint that must be respected for the solution of the kite speed within the force balance. The
value for $\kappa$ is solved numerically.

During the cyclic motion of the kite through the pattern, the apparent wind speed varies due to the aerodynamic work and the potential and kinetic energy exchange. The apparent wind velocity will be highest at the bottom and lowest on the topmost part of the pattern (Eijkelhof and Schmehl, 2022). This variation in velocity leads to oscillations of the mechanical power. Although it should ideally not affect the pattern average power during the cycle, it will demand oversizing of the drivetrain
to be able to handle the oscillation peaks. This will lead to increased costs and reduced overall efficiency since the drivetrain will not operate near its rated conditions most of the time. This undesired effect is more extreme for larger kite masses. The oscillating mechanical power must be capped if it exceeds the generator limit. This can be done in multiple ways, for example, by modulating the reeling speed or changing the angle of attack, which can both be done relatively quickly or by increasing the pattern elevation angle, which takes more time. The work of gravity during the upward and downward parts of the pattern
is conserved, but at the same time, there are non-conservative forces, such as the drag force, which lead to energy dissipation. We choose the same representative point to evaluate the mean pattern reel-out power as discussed in Sect. 2.1. The power is estimated using Eq. (25) with $\theta = \pi/2 - \beta_p$ and $\phi_p = \sin^{-1}(4\sin\gamma_p/3\pi)$.

## 2.4 Retraction phase

At the end of the reel-out phase, when the kite is at the topmost point along its trajectory, it is assumed to be pulled back
in a straight line starting from the top of the pattern, covering the reeled-out distance. This is as shown in Fig. 8, but with





the difference that the kite does not have a tangential velocity, i.e., the kite's tangential velocity component $v_{k,\tau} = 0$, with $\theta = \pi/2 - (\beta_p + \gamma_p)$ and $\phi_p = 0$. It only has a velocity in the negative radial direction. This is the reel-in velocity $v_i$, an independent variable in the model controlled by the ground station.

A force balance similar to the one described in Sect. 2.3 is solved to estimate the required mechanical reel-in power

$$P_{m,i} = F_t v_i \tag{28}$$

$$= \left( \sqrt{ \left( \frac{1}{2} \rho S \right)^2 (C_L^2 + C_D^2) \left[ (v_w \sin\theta \cos\phi + v_i)^2 + (v_w \cos\theta \cos\phi)^2 \right]^2 - (F_g \sin\theta)^2 } - F_g \cos\theta \right) v_i. \tag{29}$$

In contrast to the reel-out phase, the effect of gravity assists the kite in the retraction phase by reducing the required tether force for reeling in. When the reel-in speed is increased, the time required for reel-in can be decreased, but this increases the apparent speed, consequently increasing the tether force. To achieve this descent, the kite needs to modulate $C_L$ to a lower value by pitching the kite. By doing so, the kite could be reeled in faster without necessarily increasing the tether force, hence minimising the required reel-in power. This trade-off should be captured when optimising the system's performance.

## 2.5 Effect of vertical wind shear

Since the kite gradually climbs from lower to higher heights during the reel-out phase, it is exposed to vertical wind shear. The wind resource varies with the height from location to location based on the ground surface roughness and other local meteorological parameters (Bechtle et al., 2019). These vertical wind distributions can be modelled using meteorological data and exhibit significant diurnal and annual variations. Schelbergen et al. (2020) proposed a method to identify characteristic shapes of the wind profile using reanalysis data. Such characteristic wind profile shapes can be used with this model to evaluate the energy production of systems. Since an in-depth wind resource characterisation is not the focus of this work, the commonly used characterisation of a vertical wind profile in neutral atmospheric conditions using the power law given by

$$v_w(z_2) = v_w(z_1) \left( \frac{z_2}{z_1} \right)^\alpha , \tag{30}$$

is used to describe the relationship between wind speed $v_w$ and height $z$ based on the ground surface roughness parameter $\alpha$ (Peterson and Hennessey, 1978).

To account for the changing inflow during the system's operation, the reel-out phase is discretised in several segments as shown in Fig. 9. The tractive power is evaluated for each segment using the corresponding wind speed and resulting force balance. The orange points represent the numerical evaluation points during the reel-out and reel-in phases.

## 2.6 Electrical cycle power optimisation

The power represented by Eq. (25) is the average mechanical reel-out power over one pattern for a given wind speed. This does not yet account for the losses due to the power consumed in the reel-in phase and the losses in the drivetrain. The drivetrain is the chain of components between the drum of an AWE system and the point of connection to the electricity grid. Since the




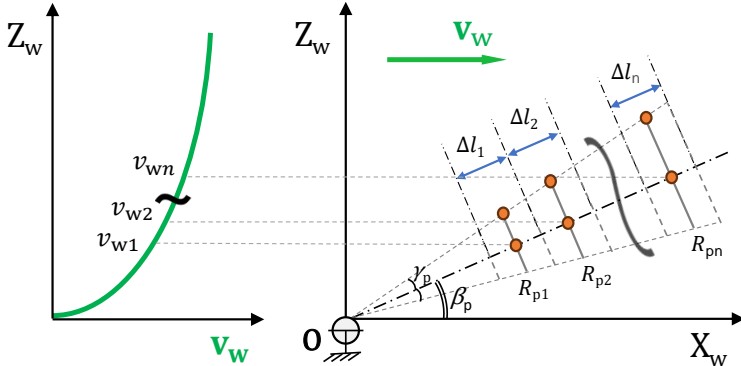

**Figure 9.** The discretized reel-out phase experiences different wind speeds as an effect of the vertical wind shear.

power output of a ground-gen AWE system is cyclic, a storage component needs to be used to charge and discharge during the cycle to maintain smooth power fed into the grid (Joshi et al., 2022). The electrical cycle average power is computed as

$$P_{\text{e,avg}} = \frac{P_{\text{e,o}}t_{\text{o}} - P_{\text{e,i}}t_{\text{i}}}{t_{\text{o}} + t_{\text{i}}}, \quad \text{where} \tag{31}$$

$$P_{\text{e,o}} = P_{\text{m,o}}\eta_{\text{DT}}, \quad \text{and} \tag{32}$$

$$P_{\text{e,i}} = \frac{P_{\text{m,i}}}{\eta_{\text{DT}}}. \tag{33}$$

Here, $t_{\text{o}}$ is the time duration of the reel-out phase, $P_{\text{e,i}}$ is the power required during reel-in, $t_{\text{i}}$ is the time duration of the reel-in phase, $\eta_{\text{DT}}$ is the drivetrain efficiency.

### 2.6.1 Drivetrain efficiency

In a typical electrical drivetrain, the generator is connected to the drum using a gearbox. The generator is then connected to an electrical storage module via a power converter and to the grid via a power converter in parallel configuration (Joshi et al., 2022; Fechner and Schmehl, 2013). Therefore, $\eta_{\text{DT}}$ is a combination of the individual component efficiencies given as

$$\eta_{\text{DT}} = \eta_{\text{gb}}\eta_{\text{gen}}\eta_{\text{pc}}\eta_{\text{sto}}\eta_{\text{pc}}, \tag{34}$$

where $\eta_{\text{gb}}$ is the gearbox, $\eta_{\text{gen}}$ is the generator, $\eta_{\text{sto}}$ is the electrical storage, and $\eta_{\text{pc}}$ is the power converter efficiencies, respectively. We assume a value of 95% for all the three components except the generator. The generator efficiency at its rated speed could be as high as 95% and drops steeply to zero below about 20% of its rated speed. This non-linear behaviour is modelled using

$$\eta_{\text{gen}} = 0.671 \left(\frac{v}{v_{\text{rated}}}\right)^3 - 1.4141 \left(\frac{v}{v_{\text{rated}}}\right)^2 + 0.9747 \left(\frac{v}{v_{\text{rated}}}\right) + 0.7233, \tag{35}$$

where $v$ is the operating speed of the generator.





### 2.6.2 Reel-out and reel-in time

The reel-out and reel-in times heavily influence the average electrical cycle power of the system. They are dependent on the
reel-out speed $v_\mathrm{o}$, reel-in speed $v_\mathrm{i}$, stroke length $\Delta l$, and the given maximum drum acceleration $a_\mathrm{d,max}$. Figure 10 shows a
velocity-time graph for a representative cycle. The reel-out phase starts with a reeling speed of zero. The kite achieves its
set reel-out speed by accelerating with $a_\mathrm{d,max}$ and remains constant until the kite covers the stroke length. The kite then
decelerates back to zero to begin the reel-in phase. Similar to the start of the reel-out phase, the reeling speed reaches its set
value by accelerating with $a_\mathrm{d,max}$ and then remains constant till the end of the reel-in phase, after which it again decelerates to
zero to begin a new cycle.

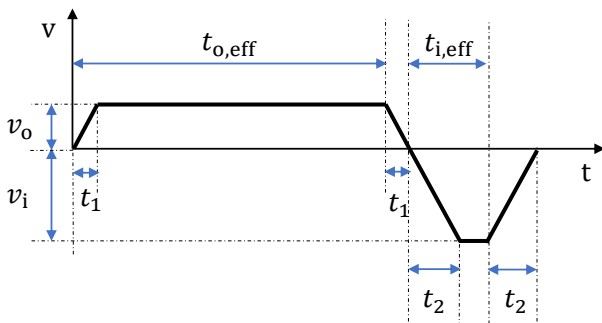

**Figure 10.** Velocity-time graph for a representative cycle.


If $t_1$ is the time taken by the kite to reach the maximum reel-out speed, $t_\mathrm{o,eff}$ is the effective time during which the kite is in
traction, and is producing power, $t_2$ is the time taken by the kite to reach its maximum reel-in speed, and $t_\mathrm{i,eff}$ is the effective
time during which the system is out of traction, then the cycle time is expressed as

$$t_\mathrm{cycle} = t_\mathrm{o} + t_\mathrm{i} = t_\mathrm{o,eff} + t_1 + t_\mathrm{i,eff} + t_2, \quad \text{i.e.} \tag{36}$$

$$t_\mathrm{cycle} = \frac{\Delta l}{v_\mathrm{o}} + \frac{v_\mathrm{o}}{a_\mathrm{d,max}} + \frac{\Delta l}{v_\mathrm{i}} + \frac{v_\mathrm{i}}{a_\mathrm{d,max}}. \tag{37}$$

### 2.6.3 Optimisation problem setup

The optimisation objective is maximising the electrical cycle average power $P_\mathrm{e,avg}$, given by Eq. (31), for given wind con-
ditions. Table 2 shows the list of the variables, and Table 3 presents the list of the constraints of the optimisation problem.
Constraints are enforced on the minimum ground clearance, required electrical rated power, peak mechanical power (limiting
the size of the generator), maximum tether length, maximum allowable tether force, and minimum number of patterns per
cycle. At least one full pattern during a cycle is imposed to account for the fact that inertial effects are excluded, and it can
be unrealistic to have fast transitions between reel-out and reel-in without completing at least one circular trajectory. Another
important constraint that must be respected during reel-out and reel-in is given by Eq. (27). Since the design space is con-
tinuous and has non-linear constraints, sequential quadratic programming (SQP), a gradient-based optimisation algorithm, is





implemented in MATLAB to solve the problem. The results give the optimal operation set-points for the defined system with
respect to the given wind conditions.

**Table 2.** Operational parameters which are optimised for given wind conditions.

| Design variable | Unit | Description |
| --- | --- | --- |
| $\Delta l$ | m | Stroke length |
| $\beta_\mathrm{p}$ | deg. | Pattern elevation angle |
| $\gamma_\mathrm{p}$ | deg. | Pattern cone opening angle |
| $R_\mathrm{p,min}$ | m | Initial turning radius |
| $v_\mathrm{o}$ | m/s | Reel-out speed |
| $C_\mathrm{L,o}$ | - | Reel-out wing lift coefficient |
| $v_\mathrm{i}$ | m/s | Reel-in speed |
| $C_\mathrm{L,i}$ | - | Reel-in wing lift coefficient |

**Table 3.** Optimisation problem constraints.

| Constraint | Unit | Description |
| --- | --- | --- |
| $h_\mathrm{min} \leq z_\mathrm{k} \leq h_\mathrm{max}$ | m | Operation height limits |
| $P_\mathrm{e,avg} \leq P_\mathrm{rated}$ | W | Electrical rated power |
| $P_\mathrm{m,o} \leq P_\mathrm{gen,rated}$ | W | Peak mechanical power |
| $l_\mathrm{t} \leq l_\mathrm{t,max}$ | m | Maximum tether length |
| $F_\mathrm{t} \leq F_\mathrm{t,max}$ | N | Maximum tether force |
| $N_\mathrm{p} \leq N_\mathrm{p,min}$ | - | Minimum number of patterns per cycle |

The optimizer attempts to find a solution for the quasi-steady force balance for every wind speed in the given range. The
cut-in wind speed $v_\mathrm{w,cut-in}$ is the minimum speed at which the system produces net positive electrical cycle power. That is, for
wind speeds below $v_\mathrm{w,cut-in}$, power will be consumed to keep the kite in the air ($P_\mathrm{e,avg} < 0$). The rated wind speed $v_\mathrm{w,rated}$
is the speed at which the system produces its nameplate-rated electrical power. Therefore, the cut-in and the rated wind speeds
are part of the solution. On the other hand, the cut-out wind speed $v_\mathrm{w,cut-out}$ is a design choice and will most probably be a
consequence of the structural lifetime and design limits of the system components. Following conventional wind turbines, the
cut-out wind speed is assumed to be $25\,\mathrm{m\,s}^{-1}$ at the operational height.





## 3    Results and Discussion

The attainable power curve for a fixed-wing ground-gen AWE system can be estimated using the presented model. Results from simulating a system with a rated power of $150\,\mathrm{kW}$ are presented in Sect. 3.1 and some effects of scaling are discussed in Sect. 3.3, showcasing the capabilities of the model and its application for the conceptual design phase.

### 3.1    Simulation results of a 150 kW system

The system's parameters are based on the prototype AP3 (Fig. 11), originally developed by Ampyx Power (Kruijff and Ruiterkamp, 2018; Vimalakanthan et al., 2018), and since 2023 continued by Mozaero (Paelink and Rand, 2024). Table 4

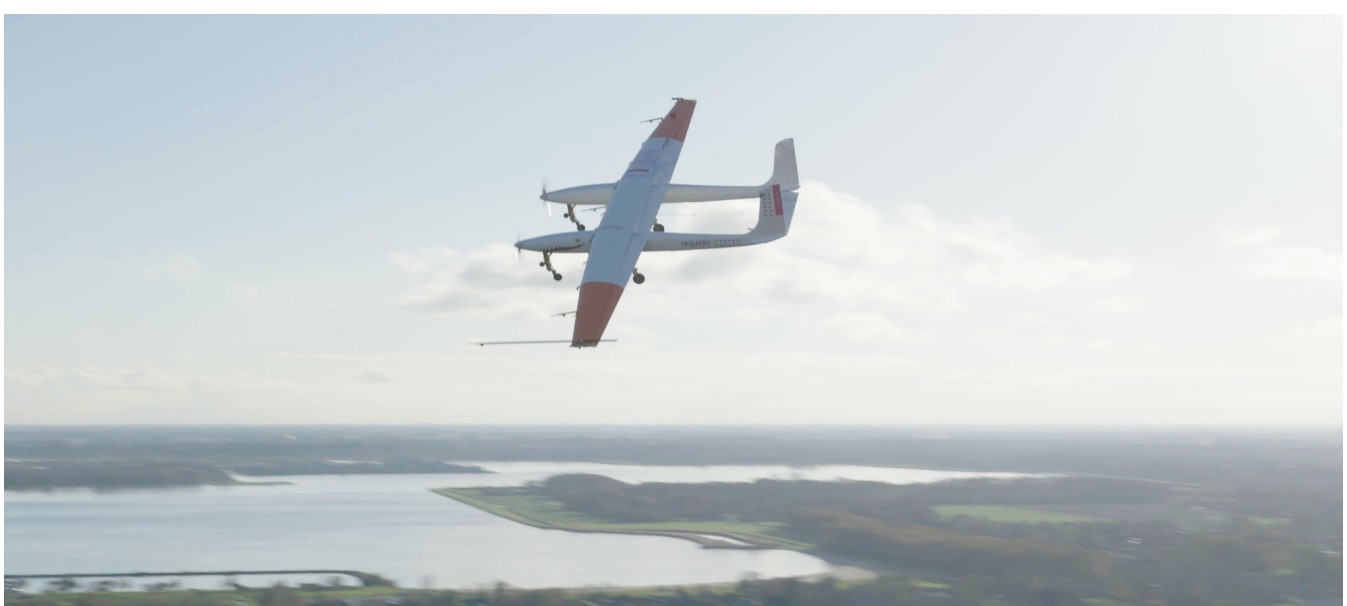

**Figure 11.** First, untethered flight of the AP3 demonstrator aircraft in the Netherlands, in November 2023 (Paelink and Rand, 2024).


lists the parameters and the limits used to define the specific system. It is important to note that the system parameters are not optimised, and hence, the power curve does not characterize a commercial $150\,\mathrm{kW}$ system. A safety factor $\eta_{\mathrm{t,gust}}$ is applied on $F_{\mathrm{t,max}}$, reducing the maximum allowable tether force value below the actual limit of the tether. To account for 3-D wing aerodynamic effects, an aerodynamic efficiency factor is applied on the maximum airfoil lift coefficient, setting an upper limit

for the wing lift coefficient $C_{\mathrm{L}}$. This is given as

$$C_{\mathrm{L,max}} = \eta_{\mathrm{C_l}} C_{\mathrm{l,max}}, \tag{38}$$

where $\eta_{\mathrm{C_l}}$ is the efficiency factor and $C_{\mathrm{l,max}}$ is the maximum airfoil lift coefficient. For the induced drag calculation using Eq. (19), a wing planform efficiency factor $e$ (Oswald efficiency factor) is used.





Figure 12 shows the chosen vertical wind shear profile representing an onshore scenario and neutral atmospheric conditions
using a surface roughness coefficient $\alpha$ of 0.143. The figure also shows wind profiles from Cabauw, an onshore location
and Ijmuiden, an offshore location in the Netherlands. These two profiles were generated using the wind profile clustering
approach described in (Schelbergen et al., 2020) and were utilised in (Eijkelhof and Schmehl, 2022). For any given location,
several profiles exist based on the probability of occurrence. The profiles with the highest probabilities in the two locations are
shown in the figure. The modelled profile with $\alpha = 0.143$ is comparable to the empirical onshore profile and hence is chosen
to represent a generic onshore location.

**Table 4.** Model input parameters list.

| Parameter | Description | Value |
|---|---|---|
| $\alpha$ | Wind shear coefficient | 0.143 |
| $S$ | Wing surface area | $12\,\mathrm{m}^2$ |
| $\mathcal{R}$ | Wing aspect ratio | 12 |
| $C_{\mathrm{l,max}}$ | Max. airfoil lift coefficient | 2.5 |
| $\eta_{\mathrm{C_l}}$ | Airfoil efficiency factor | 0.80 |
| $C_{\mathrm{l,Cd,min}}$ | Lift coefficient at minimum drag coefficient | 0.65 |
| $C_{\mathrm{d,min}}$ | Minimum drag coefficient | 0.056 |
| $e$ | Wing planform efficiency factor | 0.60 |
| $F_{\mathrm{t,max}}$ | Max. allowable tether force | 42 kN |
| $\eta_{\mathrm{t,gust}}$ | Gust margin factor | 0.90 |
| $\sigma_{\mathrm{t}}$ | Tether material strength | $7 \times 10^8\,\mathrm{N\,m^{-2}}$ |
| $\rho_{\mathrm{t}}$ | Tether material density | $980\,\mathrm{kg\,m^{-3}}$ |
| $C_{\mathrm{d,t}}$ | Cross-sectional tether drag coefficient | 1.2 |
| $l_{\mathrm{t,max}}$ | Max. tether length | $1000\,\mathrm{m}$ |
| $h_{\mathrm{min}}$ | Min. ground clearance | $100\,\mathrm{m}$ |
| $h_{\mathrm{max}}$ | Max. operating height | $1000\,\mathrm{m}$ |
| $P_{\mathrm{rated}}$ | Rated electrical power | 150 kW |
| $P_{\mathrm{gen,rated}}$ | Generated mechanical power limit | 375 kW |
| $v_{\mathrm{d,max}}$ | Max. tangential drum speed | $20\,\mathrm{m\,s^{-1}}$ |
| $a_{\mathrm{d,max}}$ | Max. tangential drum acceleration | $5\,\mathrm{m\,s^{-2}}$ |
| $N_{\mathrm{p,min}}$ | Minimum number of patterns per cycle | 1 |

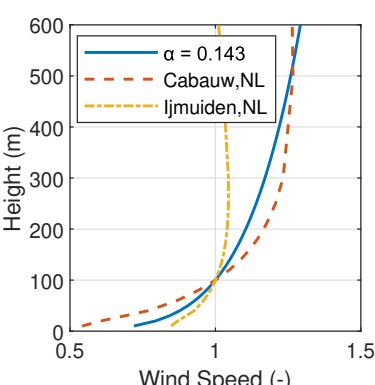

**Figure 12.** The chosen vertical wind shear profile with a surface roughness coefficient of 0.143 compared against profiles from Cabauw, an onshore location and Ijmuiden, an offshore location in the Netherlands (Eijkelhof and Schmehl, 2022).





### 3.1.1 Power curve comparison with 6DoF simulation

The six-degree-of-freedom (6-DoF) simulation results were generated using the simulation framework developed at Ampyx Power B.V., as described in Licitra et al. (2019); Ruiterkamp and Sieberling (2013); Williams et al. (2019). Figure 13 shows the computed power curve compared to the ideal crosswind power extraction theory by Loyd (1980) and the results of the 6-DoF
simulation. The horizontal axis describes the wind speed at $100\,\mathrm{m}$ height. Loyd's ideal crosswind power is computed using

$$P_{\mathrm{Loyd}} = \frac{4}{27} \frac{C_{\mathrm{L,max}}^3}{C_{\mathrm{D}}^2} \frac{1}{2} \rho S v_{\mathrm{w}}^3, \tag{39}$$

where $C_{\mathrm{L,max}}$ is the upper limit as defined by Eq. (38), $C_{\mathrm{D}}$ is computed as described in Eq. (19), and $v_{\mathrm{w}}$ is the wind speed at $100\,\mathrm{m}$ height. This ideal crosswind theory overpredicts the power because it neglects the losses due to gravity, elevation and azimuth angles, tether drag, cyclic operation, hardware limits and drivetrain efficiency.

The kite mass $m_{\mathrm{k}}$ using Eq. (20) comes out to be $437\,\mathrm{kg}$, which is close to the indication received from the company about the AP3 prototype, as seen from Fig. 6. The shape of the estimated power curve using the developed model resembles the curve generated by the 6-DoF simulations, but it is more optimistic. This is mainly because the developed model ignores the losses due to control and inertial effects. It also does not account for realistic take-off or flight sustenance conditions at low wind speeds, which is most likely the reason for the earlier cut-in. The rated power is reached at the wind speed of $15\,\mathrm{ms}^{-1}$. As a
design choice, the cut-out wind speed is chosen to be $25\,\mathrm{ms}^{-1}$ at the operational height. Due to the vertical wind shear, this translates to a wind speed of $21\,\mathrm{ms}^{-1}$ at $100\,\mathrm{m}$.

The mean mechanical and electrical, reel-out and reel-in powers, and the electrical cycle average power, are shown in Figs. 14 and 15. The reel-out power has three regimes, as described in Luchsinger (2013); Kruijff and Ruiterkamp (2018). The cubic regime I is above the cut-in speed (here, $6\,\mathrm{ms}^{-1}$), in which the reel-out power increases cubically until $10\,\mathrm{ms}^{-1}$ when the
maximum allowable tether force (here, $34\,\mathrm{kN}$, considering the gust margin factor) is reached. The linear regime II starts when reaching the maximum allowable tether force, in which the reel-out power increases linearly till the chosen rated electrical power (here, $150\,\mathrm{kW}$) is reached. The flat regime III starts when reaching the rated power and continues till the cut-out speed. In this regime, the mechanical reel-out power is capped to maintain the rated electrical power. The power is capped by varying the operational parameters. These changes in operational parameters also affect the reel-in power seen in Fig. 15.

### 3.1.2 Forces and operational parameters for the entire wind speed range

Figures 16 and 17 show the resultant aerodynamic force, the tether force and the gravitational force during the reel-out and reel-in phases, respectively. As specified in Table 4, a gust margin factor of 0.9 is applied to the maximum allowable tether force. Once this upper limit is reached, the aerodynamic force has to be capped to avoid tether overload. In our specific example, this limit is reached at $10\,\mathrm{ms}^{-1}$. The aerodynamic force can be capped by reducing the wing's lift coefficient, modulating the
reeling speed, or increasing the elevation angle. The choice of a specific capping strategy depends on multiple trade-offs. The optimisation objective is the average electrical cycle power, including the reel-out and reel-in phases. During reel-in, the kite is flown such that the maximum contribution of the generated aerodynamic force is used to counter the kite's weight, reducing

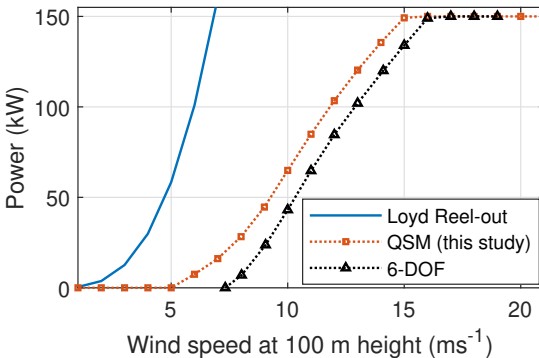

**Figure 13.** Power curve comparison of the QSM with Loyd and 6-DoF simulation results.

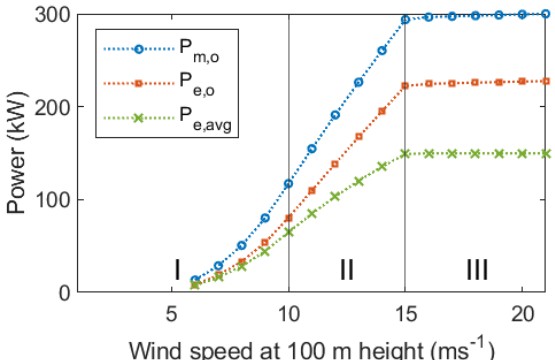

**Figure 14.** Mean mechanical power $P_{m,o}$, electrical reel-out power $P_{e,o}$, and electrical cycle average power $P_{e,avg}$ as functions of the wind speed.

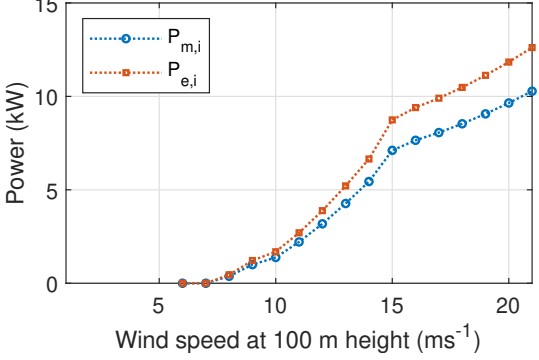

**Figure 15.** Mean mechanical reel-in power $P_{m,i}$ and electrical power $P_{e,i}$ as functions of the wind speed.

the required pulling force and, consequently, the reel-in power. This is seen in Fig. 17. Intuitively, if the aerodynamic force completely balanced the weight, it would lead to a $F_{t,i} = 0$ and hence no requirement of reel-in energy. This would be the case

of a freely gliding kite. However, this could also increase the reel-in time, which could lead to lower net cycle power. Hence, the optimizer finds a solution to the reel-in speed such that it creates a non-zero tether force but still ultimately reduces the net cycle loss.

Figures 18 and 19 shows the lift and drag coefficients during the reel-out and reel-in phases. As stated earlier, $C_L$ is a variable in our model and $C_D$ is calculated using Eq. (17). The aerodynamic force during reel-in only has to counter the kite's weight,

which is achieved by decreasing the lift coefficient during reel-in. Because of the lift-induced drag contribution, the kite drag coefficient is a function of the lift coefficient and follows its trend. The total drag coefficient is the summation of the kite drag coefficient and the tether drag coefficient as described in Eq. (17). Figure 19 shows that the tether drag coefficient contributes





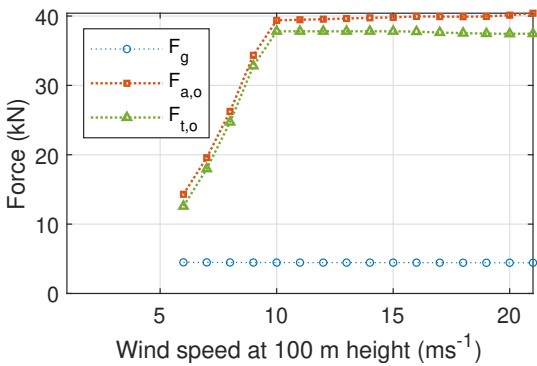

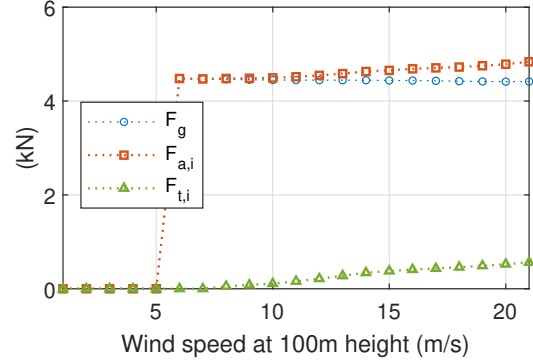

**Figure 16.** Mean resultant aerodynamic force $F_{a,o}$, tether force $F_{t,o}$, and weight of the kite and the tether lumped together $F_g$ during the reel-out phase, as functions of the wind speed.

**Figure 17.** Mean resultant aerodynamic force $F_{a,i}$, tether force $F_{t,i}$, and weight of the kite and the tether lumped together $F_g$ during the reel-in phase, as functions of the wind speed.

significantly to the system's total drag coefficient. It is almost equal to the kite drag coefficient during the reel-out phase and is higher during the reel-in phase.

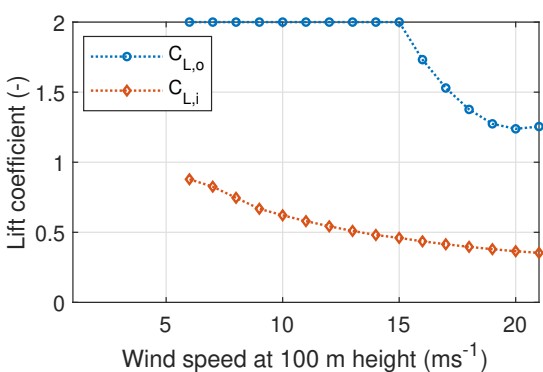

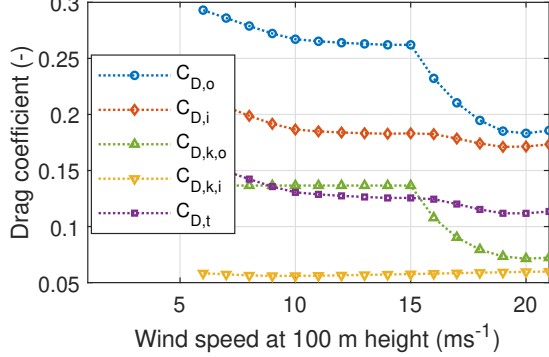

**Figure 18.** Mean kite lift coefficients $C_{L,o}$ and $C_{L,i}$ during the reel-out and reel-in, respectively, as functions of the wind speed.

**Figure 19.** Mean effective system drag coefficients $C_{D,o}$ and $C_{D,i}$, mean kite drag coefficients $C_{D,k,o}$ and $C_{D,k,i}$ during the reel-out and reel-in, respectively, and the mean tether drag coefficient $C_{D,t}$, as functions of the wind speed.

The kite's radial and tangential velocity components are commonly non-dimensionalised with the wind speed, leading to the reeling factor $f = v_{k,r}/v_w$, and the tangential velocity factor $\lambda = v_{k,\tau}/v_w$ (Schmehl et al., 2013). Figure 20 shows the tether reeling factors during reel-out and reel-in and the tangential velocity factor during reel-out. The reel-out speed peaks when the rated power is reached, i.e. at $15\,\mathrm{m\,s^{-1}}$ of wind speed, and then gradually reduces, assisting in power capping. The reel-in speed is kept at the drum's tangential speed limit of $20\,\mathrm{m\,s^{-1}}$. This is seen from the gradual decrease of the reel-in speed





factor. After the maximum tether force is reached at $10\,\mathrm{m\,s^{-1}}$, the kite's tangential velocity is gradually reduced, decreasing the aerodynamic force to maintain the tether force at its maximum value.

Figure 21 shows the reel-out time, reel-in time, average time the kite takes to perform one circular pattern during reel-out, and the number of patterns per cycle. The number of patterns is calculated using the reel-out time, pattern radius and tangential kite speed as

$$N_{\mathrm{p}} = \frac{t_{\mathrm{o}}}{2\pi R_{\mathrm{p}}/(v_{\mathrm{k},\tau})}. \tag{40}$$

In a more realistic operation, the number of patterns should be a whole number such that the reel-in phase always starts from the top point of the pattern. However, since we are not resolving the full trajectory in this model, the number of patterns is allowed to be a fractional result. Moreover, since inertial effects are ignored in this model, the full cycle time durations are optimistic. Realistic cycle times will increase due to the transition phase between reel-out and reel-in, which is unaccounted

for in this model.

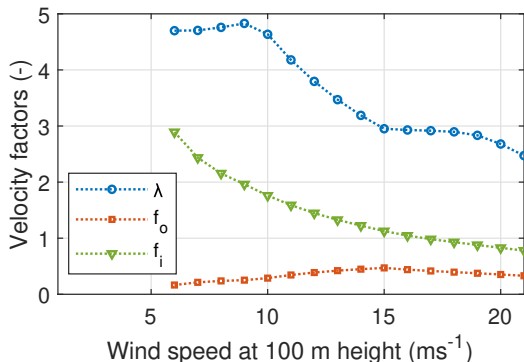

**Figure 20.** Mean kite tangential speed factor $\lambda$, reel-out factor $f_{\mathrm{o}}$, and reel-in factor $f_{\mathrm{i}}$, as functions of the wind speed.

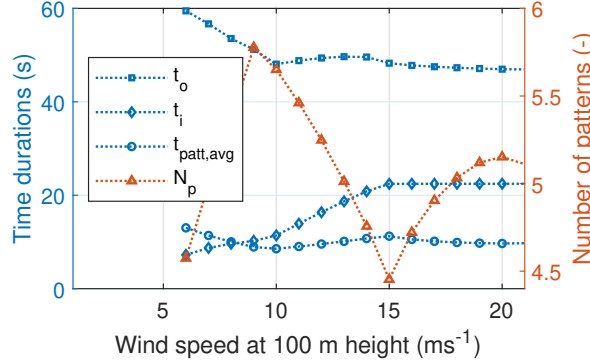

**Figure 21.** Reel-out time $t_{\mathrm{o}}$, reel-in time $t_{\mathrm{i}}$, average pattern time $t_{\mathrm{patt,avg}}$, and number of patterns per cycle $N_{\mathrm{p}}$, as functions of the wind speed.

Figure 22 shows the average pattern height, pattern radius, stroke length, maximum tether length, and minimum tether length. The average pattern height is the height of the centre point of the pattern at half of the stroke length. The minimum tether length and, consequently, the pattern radius and height are primarily driven by the ground clearance constraint, pattern elevation angle and the cone opening angle. In reality, they will also be influenced by the effect of the centrifugal force, which

is ignored in the quasi-steady approach. As the elevation angle increases, the required minimum tether length is reduced. The maximum tether length is driven by the optimised stroke length.

Figure 23 shows the roll, pattern elevation, and opening cone angles. The roll angle is the deviation of the resultant aerodynamic force vector with respect to the radial direction. The pattern elevation angle increases with the wind speed. This quasi-steady flight state results from the trade-off between the increase in incoming wind speed, an increase in cosine losses




due to gravity and a decrease in reel-in power with a higher elevation angle. The optimizer trades all these factors to maximise the average cycle power at each wind speed.

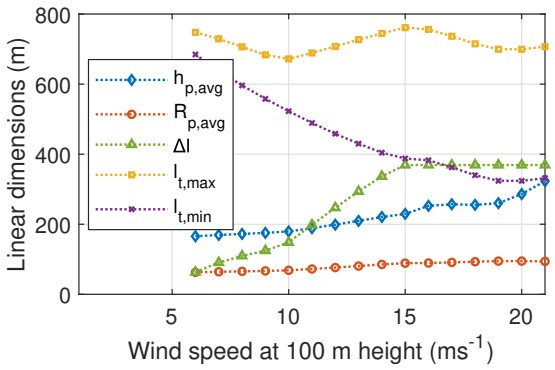

**Figure 22.** Average pattern height $h_{\mathrm{p,avg}}$, average pattern radius $R_{\mathrm{p,avg}}$, stroke length $\Delta l$, maximum tether length $l_{\mathrm{t,max}}$, and minimum tether length $l_{\mathrm{t,min}}$, as functions of the wind speed.

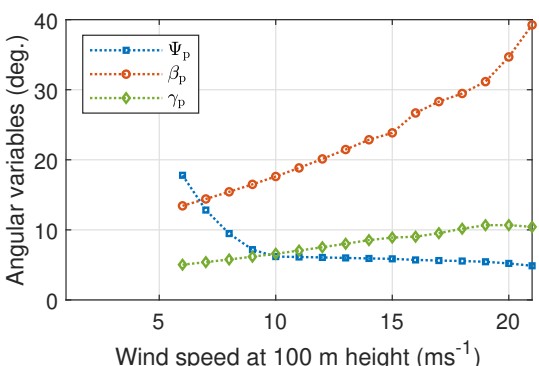

**Figure 23.** Mean roll angle $\Psi_{\mathrm{p}}$, average pattern elevation angle $\beta_{\mathrm{p}}$, and opening cone angle $\gamma_{\mathrm{p}}$, as functions of the wind speed.

### 3.1.3 Forces and operational parameters over one cycle

The maximum convertable power is limited by the generator-rated power, which in our specific example is $375\,\mathrm{kW}$, as given in Table 4. To enforce this hardware limit in the third wind speed regime, the operational parameters have to be modulated.

Figure 24 shows the mechanical, electrical and electrical cycle power over a single pumping cycle at the rated wind speed of $15\,\mathrm{m\,s^{-1}}$. The delivered rated power of $150\,\mathrm{kW}$ is the electrical cycle average power. The difference in instantaneous mechanical and electrical power is due to the drivetrain efficiency.

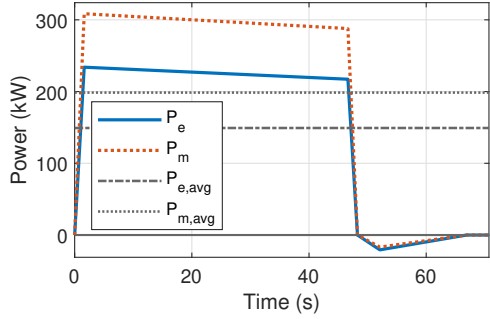

**Figure 24.** Instantaneous powers $P_{\mathrm{e}}$ and $P_{\mathrm{m}}$ together with net powers $P_{\mathrm{e,avg}}$ and $P_{\mathrm{m,avg}}$ over one pumping cycle at rated wind speed of $15\,\mathrm{m\,s^{-1}}$.





The power profile during the cycle has a slight downward trend during the reel-out and an upward trend during the reel-in. This is explained using Fig. 25. For the quasi-steady state evaluation, the reel-out phase is discretized into five segments

arranged in sequence on the horizontal axes of the diagrams. The cycle begins with a tether length of around $400\,\mathrm{m}$, and the reel-out phase ends with a tether length of around $700\,\mathrm{m}$. The average pattern height and the pattern radius increase during the reel-out phase. Due to the gain in height, the kite experiences a higher wind speed $v_\mathrm{w}$ as it climbs up. Due to the increasing tether length, the overall drag of the system increases, and hence, the glide ratio decreases. Hence, to respect the relation given by Eq. (27), the kite speed has to drop, reflected in the reeling and tangential velocity factors. Since the tether force is already

maximised, the overall power decreases due to a lower reel-out speed.

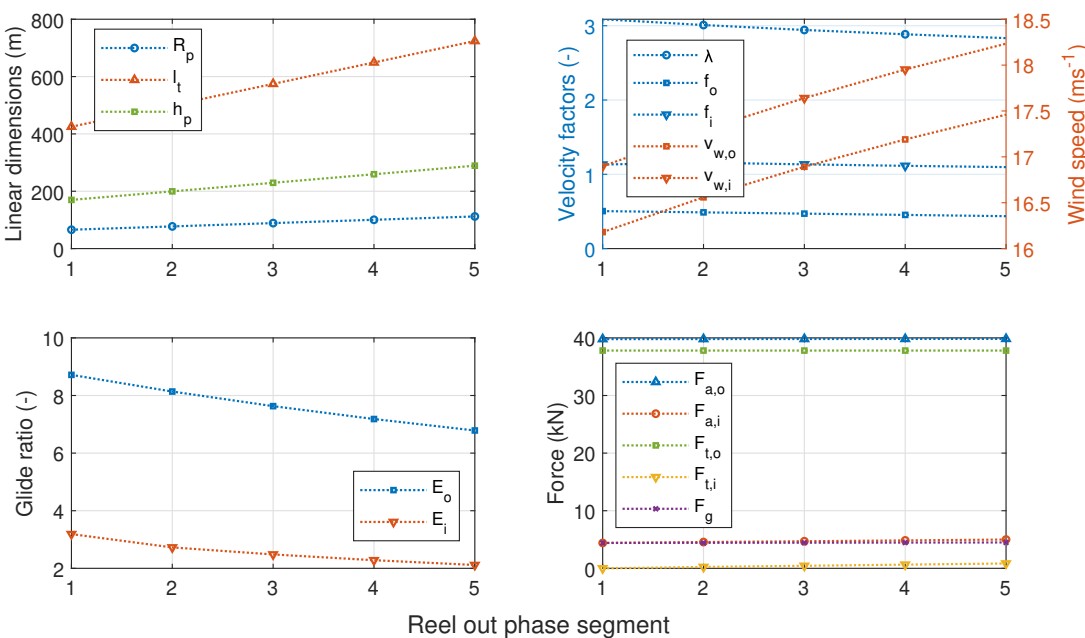

**Figure 25.** Evolution of parameters over the discretized reel-out phase in five segments at rated wind speed of $15\,\mathrm{m\,s^{-1}}$.

## 3.2 Effect of gravity

Figure 26 shows the power curve comparison between two simulations, one including and one without including the effect of gravity (i.e. weight). Gravity has a negative impact on operation at low wind speeds because this affects the attainable reel-out power substantially. Hence, the simulation without gravity yields better performance at lower wind speeds than the

one including gravity. But for higher wind speeds, this effect is superseded by its impact on the reel-in phase. As explained in Sect. 2.4, the weight assists in the retraction phase, positively impacting the net cycle power output. The kite is pitched in such a way that the resultant aerodynamic force vector balances the gravitational force vector, hence reducing the tether force magnitude in the quasi-steady force balance. The kite can be retracted faster without consuming a lot of energy.



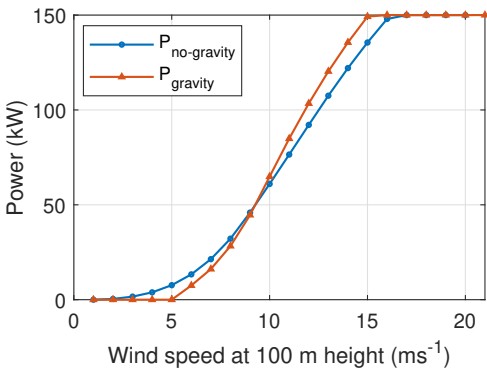

**Figure 26.** Power curve comparison with and without the effect of gravity.

Figures 27 and 28 detail the effect of gravity over a pumping cycle for a lower wind speed of $6\,\mathrm{m\,s^{-1}}$ and the rated wind
speed $15\,\mathrm{m\,s^{-1}}$. The reel-out power without the effect of gravity is substantially higher at $6\,\mathrm{m\,s^{-1}}$ than at $15\,\mathrm{m\,s^{-1}}$. The net
difference between the energy generated during reel-out and consumed during reel-in leads to higher net average power for the
case without gravity at $6\,\mathrm{m\,s^{-1}}$ than at $15\,\mathrm{m\,s^{-1}}$. This shows that excluding gravity in the analysis does not necessarily lead to
optimistic results. In any case, including gravity should always be the more realistic simulation for the pumping cycle.

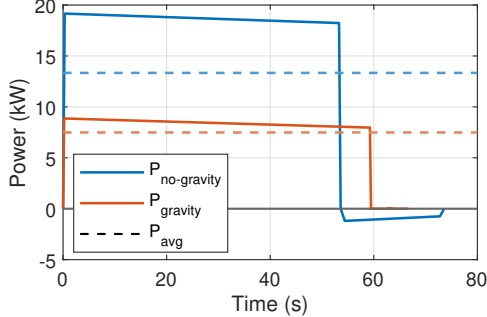

**Figure 27.** Cycle power comparison with and without the effect of gravity for low wind speed of $6\,\mathrm{m\,s^{-1}}$.

**Figure 28.** Cycle power comparison with and without the effect of gravity for higher wind speed of $15\,\mathrm{m\,s^{-1}}$.

### 3.3 Effect of scaling

One of the primary purposes of this model is to capture the effects of scaling on the performance of fixed-wing AWE systems.
Due to the interdisciplinary nature of AWE systems, multiple trade-offs must be considered. The kite and the tether are the
primary aspects affecting the system's performance. The performance metric used is the power harvesting factor $\zeta$ defined as

$$\zeta = \frac{P}{P_{\mathrm{w}} S}, \tag{41}$$





where $P$ is the extracted mechanical power and $P_\mathrm{w}S$ is the available power in the wind. This metric is based only on the reel-out power and does not consider the reel-in power. The force a tether can withstand for a given material strength is proportional to its diameter as shown in Eq. (12). The kite should also be able to withstand this tether force; hence, with increasing tether force, the structural mass of the kite increases to support the increasing wing loading. Though increasing the tether force will enable the extraction of more power, the consequent increase in kite mass will decrease the performance. Also, the contribution of tether drag will increase with increasing diameter, consequently penalising the extractable power.

Figure 29 shows the effect of tether diameter on the performance of an AWE system. For a kite with the same wing area, the mass increases with increasing tether force as given in Eq. (20). This increase in kite mass negatively impacts the attainable reel-out power. For this simulation, the performance is maximum when using a tether of diameter of around $3.8\,\mathrm{cm}$. Similarly, Fig. 30 shows the effect of scaling the wing area on the performance of an AWE system. For the chosen tether, the kite wing area which maximises power is $50\,\mathrm{m}^2$.

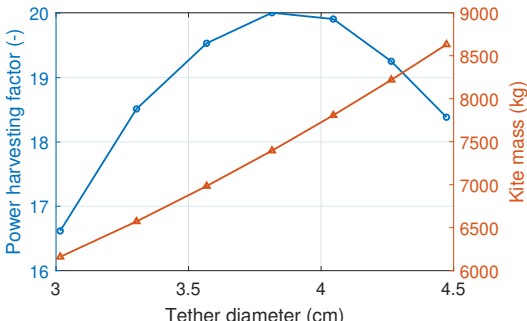

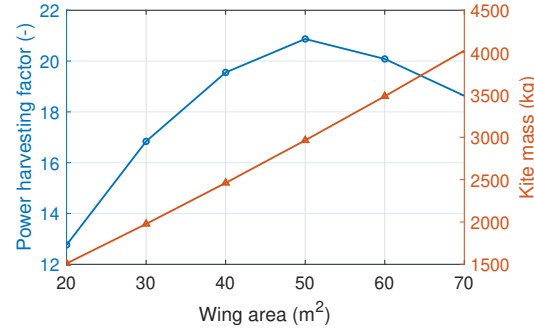

**Figure 29.** Effect of tether diameter on the performance of an AWE system with a fixed kite wing area of $100\,\mathrm{m}^2$ at a constant wind speed of $12\,\mathrm{m\,s}^{-1}$.

**Figure 30.** Effect of wing area on the performance of a system with fixed tether diameter of $2.7\,\mathrm{cm}$ at a wind speed of $12\,\mathrm{m\,s}^{-1}$.

Figure 31 shows the effect of the tether diameter on the performance of a system with a fixed kite wing area of $100\,\mathrm{m}^2$ for the complete operational wind speed range. As seen from Fig. 29, the kite mass of a system with smaller tether tension is lower. Lighter kites will experience lower gravitational loss and, hence, will perform better at lower wind speeds. At higher wind speeds, the maximum tether force limits the extractable power. Therefore, for a given wing area, systems with thinner tethers, i.e. lower $F_\mathrm{t,max}$, perform better at lower wind speeds, and systems with larger tethers, i.e. higher $F_\mathrm{t,max}$, perform better at higher wind speeds.

Figure 32 shows the wing area's effect on a system's performance with a fixed wing-loading for the complete operational wind speed range. Fixed wing loading is used instead of a fixed tether force since simulation results of a high tether force coupled to a small kite and vice-versa do not converge for the entire operational wind speed range. A larger tether force demands a stronger kite, resulting in a heavier one. This configuration cannot produce positive net cycle power at low wind speeds. Therefore, the choice of tether force for a given wing area must fall in a certain range to have converged results for





the entire operational wind speed range. The system performs better with increasing wing area, but these gains are diminishing since the penalising effect of the gravitational force scales faster than the performance gain.

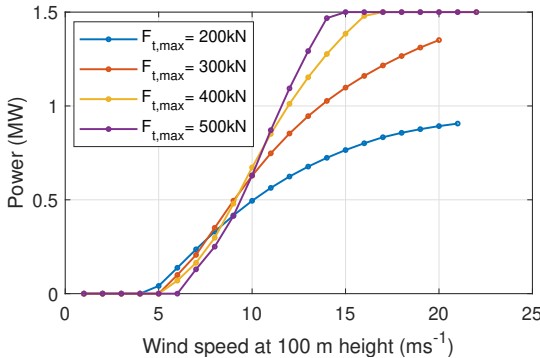

**Figure 31.** Effect of scaling the tether (diameter) on the performance of a system with a fixed kite wing area of $100\,\mathrm{m}^2$ for the complete operational wind speed range.

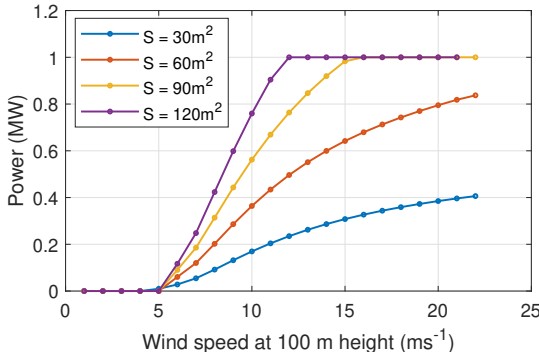

**Figure 32.** Effect of scaling the wing area on the performance of a system with a fixed wing-loading of $3\,\mathrm{kN\,m}^{-2}$ the complete operational wind speed range.

## 3.4 Discussions

The results show that the proposed model captures all relevant dependencies between the system components, allowing the
evaluation of different trade-offs at play. Since the model is based on a quasi-steady flight motion, the results are expected to be optimistic predictions of a real system's performance. Since the model relies on a limited set of input parameters defining an AWE system, it is suitable for coupling with similar-fidelity cost models, as proposed in Joshi and Trevisi (2024). The present modelling approach does not account for the various effects of inertia responsible for losses in the different operational phases. For example, the transition time between the reel-out and reel-in phases and the losses due to the centrifugal force when flying
circular manoeuvres are not considered. Eijkelhof and Schmehl (2022) found that the kite needs to be slowed down at the start and the end of the transition phase to avoid tether rupture due to the change in the magnitude of forces. This effect is also known as the 'whiplash effect'. The present model cannot estimate the reel-out power oscillations due to the acceleration and deceleration of the kite when it follows the prescribed flight path.

The model and simulation results in this paper are not validated against measurement data but compared with simulation
results of a 6-DoF dynamic model. Skysails Power Gmbh, a German company, released a certified power curve of their PN-14 system based upon the standard IEC 61400-12-1 used for conventional wind turbines (Bartsch et al., 2024). They reported good agreement of their measurements with their simulation results. Figure 33 shows an overlay of their measurements against the $150\,\mathrm{kW}$ simulation results from Sect. 3.1. This is not a performance comparison since the technologies and their system characteristics differ significantly, but the systems do have similar power ratings. The IEC standard requires multiple changes
considering the differences between conventional wind turbines and AWE systems. For instance, the definition of the reference height, wind range, method of averaging over time, incorporating the number of cycles in averaging, etc. The reference height





proposed by Skysails is $200\,\text{m}$ which is closer to the average operational height of their system. Alignment on the reference height for wind speed measurements while communicating the power curve is essential for fair comparisons. As the AWE sector advances rapidly, there is a need for dedicated IEC standards to validate the power curve of AWE systems.

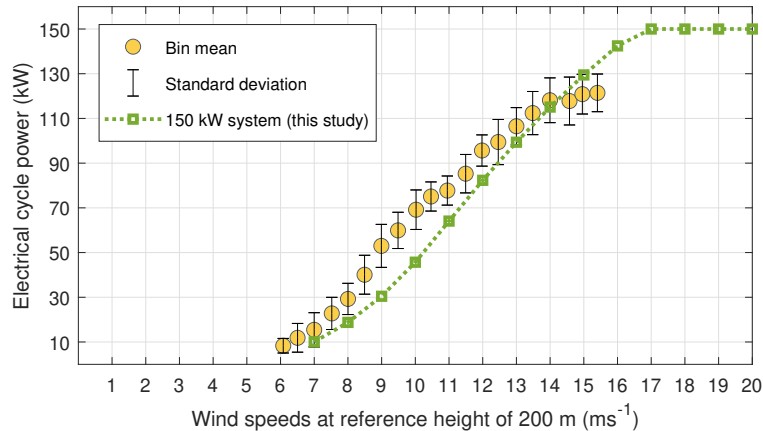

**Figure 33.** Overlay of the power curve of the $150\,\text{kW}$ system over Skysails' validated power curve of SKS PN-14 (Bartsch et al., 2024).

## 4  Conclusions

The quasi-steady model presented in this paper enables fast power curve computations based on a limited set of input parameters. It is useful for understanding the fundamental physical behaviour of fixed-wing ground-generation AWE systems and is suitable for sensitivity analysis and estimating AWE systems' theoretical potential. The model can easily be coupled to systems engineering tools, cost models, and larger-scale energy system models. It may thus help to create technology development road-maps, investigate the scaling potential, and define research targets to validate assumptions.

The kite mass is a key parameter influencing the performance of systems, primarily at lower wind speeds. A higher mass leads to a larger component of the generated aerodynamic force needed to compensate for the gravitational force, reducing the usable mechanical power. On the other hand, gravity positively impacts performance at higher wind speeds by reducing the required energy during reel-in. The tether diameter and the kite's structural mass are coupled to design an optimised system. The maximum force-bearing capacity of the tether is directly proportional to the diameter of the tether, and a higher tether force requires a structurally stronger kite. Hence, choosing a tether with a larger force-bearing capacity also increases the kite mass, negatively impacting the low wind speed performance but enabling higher power extraction at higher wind speeds. This trade-off becomes critical for choosing the optimal tether-kite combination based on site-specific requirements. Integrating the prescribed model in an optimisation framework provides a computational design tool that accounts for the multiple trade-offs for site-specific design. The system design parameters, such as the kite wing area, generator rating, tether diameter, etc., can be optimised to maximise the annual energy production for a specific wind resource. Moreover, annual energy prediction alone





will not give the right indication for system design since this metric lacks the influence of costs. To include this important aspect, the presented model can be coupled to a cost model to find the system design that minimises the levelised cost of energy.

Since the presented model is based on the assumption of quasi-steady flight motion, it does not account for inertial effects. These will be significant for larger AWE systems; hence, the model is likely too optimistic in estimating their performance. This approach occupies a middle ground between ideal power extraction and fully resolved dynamic simulations. The outcomes of this analysis define the upper limits that practical systems might approach. Consequently, these models are valuable for determining whether and under what conditions AWE could benefit the entire energy system.

*Code availability.* The model is implemented in MATLAB and is available on GitHub at: https://github.com/awegroup/AWE-Power. The repository contains a pre-defined input file which can be used to run the model and reproduce the results presented in the paper.

## Appendix: Nomenclature

### Greek symbols

$\alpha$        Wind shear coefficient

$\beta$        Elevation angle

$\chi$        Course angle

$\eta$        Efficiency

$\gamma$        Cone opening angle

$\lambda$        Tangential velocity factor

$\phi$        Azimuth angle

$\Psi$        Roll angle

$\rho$        Material density

$\sigma$        Material strength

$\tau$        Tangential

$\theta$        Polar angle

$\zeta$        Power harvesting factor

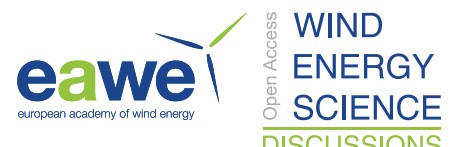

**Latin symbols**

| | |
|---|---|
| $A\!R$ | Aspect ratio |
| $a$ | Acceleration |
| $C_\mathrm{D}$ | Drag coefficient |
| $C_\mathrm{L}$ | Lift coefficient |
| $D$ | Drag |
| $d$ | Diameter |
| $e$ | Wing planform efficiency factor |
| $F$ | Force |
| $f$ | Reel-out factor |
| $h$ | Height |
| $L$ | Lift |
| $l$ | Length |
| $m$ | Mass |
| $N$ | Number |
| $P$ | Power |
| $R$ | Radius |
| $S$ | Wing area |
| $t$ | Time |
| $v$ | Velocity |
| $z$ | Z-axis co-ordinate |

**Subscripts**

| | |
|---|---|
| a | Apparent |
| avg | Average |



| | |
|---|---|
| DT | Drivetrain |
| e | Electrical |
| eff | Effective |
| g | Gravity |
| gb | Gearbox |
| gen | Generator |
| i | Reel-in |
| k | Kite |
| m | Mechanical |
| max | Maximum |
| min | Minimum |
| o | Reel-out |
| p | Pattern |
| pc | Power converters |
| r | Radial |
| ref | Reference |
| sto | Storage |
| t | Tether |
| w | Wind |




*Author contributions.* Conceptualisation, R.J., M.K. and R.S.; methodology, R.J., M.K. and R.S.; software, R.J.; validation, R.J. and M.K.; writing—original draft preparation, R.J. and M.K.; writing—review and editing, R.S.; supervision, M.K. and R.S.; funding acquisition, M.K. and R.S.

*Competing interests.* At least one of the (co-)authors is a member of the editorial board of Wind Energy Science.



*Acknowledgements.* This work is part of the NEON research program and has received funding from the Dutch Research Council NWO

under grant agreement No. 17628 and the Interreg North West Europe project MegaAWE. The authors also thank Ampyx Power B.V. for

providing reference data and inputs.



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
