# Peer review of "Power curve modelling and scaling of fixed-wing ground-generation airborne wind energy systems"

_Wind Energy Science, 2024_

## Author Comment (AC2)

**Round 1**

**Authors' response to Reviewer 1 comments**

Rishikesh Joshi

September 11, 2024

We appreciate your feedback and comments on our manuscript. Following are our responses.

The provided comments are in the standard black font, and our responses to the comments are in blue. The associated changes are in the revised manuscript submitted with this document.

**Provided comments**

1. The abstract suggests that this paper simply showcases the capability of the model by running a number different of analyses. This approach of just showing what the model can do is reflected throughout the paper. On the other hand, the results presented do have valuable insights that can be of interests to other researchers. A few examples: excluding gravity does not always give more optimistic results, optimal reel in is achieved by balancing the aerodynamic forces with gravity, thin tethers perform better at low wind speed and thick tethers perform at high wind speeds, etc. In my opinion, changing the focus of the paper to highlighting these new insights (noting that they were derived from the new model presented) would make the paper more interesting.
We understand we might not have highlighted the key insights from our paper along with the modelling framework. The insights derived using the model were only present in the Results section and were summarized in the Conclusions. We have now added text highlighting interesting insights in the Abstract and at the end of the Introduction section as well.

2. In the background section, lines 63-98 could be split into smaller paragraphs. I can see that the authors are describing how this paper improves on previous formulations. Since this cover a wide range of topics (model fidelity, gravity effects, etc.). Splitting them into different paragraphs, with each focusing on how improvements are required could further highlight the value of this paper.
We have now split the single large paragraph into smaller paragraphs focusing on particular aspects, starting from the required fidelity, the inclusion of gravity, comparison with measurement or higher fidelity simulation data, and followed by our proposed model introduction. Unfortunately, all the discussed papers will have some overlap in their approaches, and hence, a very clear categorisation is a bit difficult. We have also incorporated here the community comment by Maximilian Ranneberg.

3. Lines 132-133 ('the wind vector vw is orthogonal to the kite's tangential motion component'): I don't think Figure 4a shows that vw is orthogonal to the tangential motion.
We have now added the explanation that the circular trajectory is symmetric around the $\mathbf{X}_\mathrm{w}$ axis and changed the wording from 'situation' to 'instance'. Since the kite is at the top point of its circular trajectory, the tangential motion is out of the plane, i.e. in the $e_\phi$ direction, which is orthogonal to $\mathbf{v}_\mathrm{w}$. This is true for that instance but indeed cannot be shown in Fig 4(a) since it shows the $\mathbf{X}_\mathrm{w}\mathbf{Z}_\mathrm{w}$ plane.

4. Line 240 ('It states the maximum force it can handle per unit wing area'): repeated use of 'it'.
Changed the text by combining two sentences to avoid repetition of 'it'.

5. Line 265 (equation 23): this is quite an important equation. Is it feasible to show the derivation immediately below or in the appendix? I guess this is related to doing a simple integration with respect to dl, then relate the l*g*cos(beta) terms to the tether area and mass. I imagine that equations like this will be adopted by many preliminary studies in the future, hence the need to show the working.
The intermediate derivation steps are now added.

6. Line 307: suggest removing 'the effect of'.
   Done.

7. Line 346 (equation 35): it would be helpful to plot this relationship on a graph.
   Added the plot of the relationship.

8. Please move the figures to the relevant sections in the paper. One example: figs. 14 and 15 should be placed within section 3.1.1.
   We had already placed them in section 3.1.1, but the Overleaf formatting automatically pushes the images into the next section to minimize the white spacing in the paper. In the final formatting phase, we will convey this formatting comment to the journal editors.

9. Please check the y-axis label in figure 17.
   Corrected.

10. Line 487-488: suggest rewriting to 'one without the effect of gravity' (i.e., removing 'including').
    Changed.

---

## Author Comment (AC3)

**Round 1**

**Authors' response to Reviewer 2 comments**

**Rishikesh Joshi**

**September 11, 2024**

We appreciate your feedback and comments on our manuscript. Following are our responses.

The provided comments are in the standard black font, and our responses to the comments are in blue. The associated changes are in the revised manuscript submitted with this document.

**1 General comments**

1. The article is rather well written and structured. It deals with a very relevant subject that is not so easy to define. Basically, the aim is to develop a tool or, more precisely, a framework of tools for optimising and sizing the design of fixed-wing, ground-generated airborne wind energy systems. If one type of these keywords into Scholar, you will come across the Sommerfeld et al 2022 article, which is cited in references. The novelty of the present study is to propose a simpler "quasi-steady" model to avoid the difficulty of the kite control, which is very influential on performance according to the Sommerfeld conclusions. The modelling framework proposed in this paper is therefore better adapted to the first stages of the design. The correlative question, which I don't think I've seen addressed here, is the following: Are the trajectories proposed in this study really feasible for a real servo-controlled system? Added explanatory text regarding feasibility in terms of minimum turning radius in Section 2.6.3.

2. I also think that the title of this document could be replaced by Sommerfeld's and vice versa. The title of this study should then be changed to emphasise the novelty compared with Sommerfeld's study. As you correctly summarised, this model is aimed at preliminary or conceptual design studies. We propose a faster method for power curve modelling that allows us to understand scaling effects without the need for an implemented controller. While we attempted to expand the title to include additional aspects, doing so made the title too complex for a quick read. We now have changed the word 'Performance' to 'Power curve' because it is more specific to the goal of the model which is absent in Sommerfeld's title. This should distinguish our work from Sommerfeld's since typically, models with implemented controllers serve purposes different from power curve modelling.

**2 Specific comments**

1. The second section describes the system under study and the parameters of the flight configuration. Simplification assumptions are also set out. A simple point-mass model is used. An optimisation problem where the net electrical cycle power is maximised for given wind conditions. An optimisation problem is set, in which the net electrical power of the cycle is maximised for given wind conditions. Finally, the list of influential parameters to be studied is given. Subsection 2.1 presents an initial highly simplified model based on the zero-mass hypothesis in a truly clear and didactic manner. Highly technical considerations are considered, such as minimum and maximum flight altitudes in terms of safety, as an example. The assumptions are sometimes very approximate, as is the case for the tether's contribution to aerodynamic drag.

   Section 2.2 proposes an original model for estimating kite mass. Which is presented as a key novelty of the present study.

   Section 3 shows some results for specific applications. The first is a 150-kW system based on the 150 kW AP3 prototype developed by Ampyx Power. This case has the advantage of having been widely documented and referenced in the scientific literature. With this dataset, the simplified model developed in the current study is compared

to results obtained using the 6-DoF simulation framework developed by Ampyx. The Ampyx model is presented as a high-fidelity class. This comparison demonstrates that the current work represents a significant improvement compared to the basic zero mass model from (Loyd 1980). Hence the sensitivity to various design parameters and environmental variables such as gravity are investigated. The proposed model seems to give sensible results. Comparison with a more sophisticated model (the Ampyx model) also provides an estimate of the accuracy of the present model. Various parameter optimizations studies are then proposed to explore the ability of this model. The results analysis seems relevant and logical in terms of the influence of the various physical parameters involved (wing surface, tether diameter, weight, etc.). The section concludes with a discussion that does not overlook the limitations of the proposed model.

We agree with the reviewer's summary.

2. However, the reader misses details to really appreciate the difference between the so-called high-fidelity method by Ampyx and the present study model. The authors should provide more details on the differences between the two codes.

A description regarding the higher-fidelity model is now added at the start of Section 3.1.1.

**3 Technical corrections**

1. L12-13 In the abstract, we read several times ".., and...". I feel the commas could be deleted.

We follow British English as a writing style, and the comma in such listings is known as the 'Oxford comma'. Due to this preference, we decided to keep the commas.

2. The language is sometimes too technical, which makes for tedious reading. The full names of technical variables should be preferred to symbols in the text, as in the sentences on lines 272 and 273 by way of example. This would make for smoother reading.

We acknowledge your preference, but we have also experienced other preferences where people like to read out the names of the variables, at least in the first appearance of any section within the manuscript. Since we also share the same writing preference, we have decided to maintain our writing style regarding this.

3. L265 Equation (23), which is presented as original in the paper, should be referred to (Houska and Diehl 2006) who first proposed it.

Houska and Diehl indeed used the same result in 2006, but without providing the derivation and the assumptions necessary to arrive at the result. We have now referred to their paper in the text.

4. L 346: The reference from which equation (35) is taken is not given. Please specify.

We got this information from our industrial partner Ampyx Power B.V. Added an explanation in the text and also added a graph visualising the relationship.

5. L432 QSM acronym is used in Figure 13 legend and caption but is not explain in the text nor in the Nomenclature

Added in the Introduction section and the figure caption.

6. L530-535 Several places in the text refer to losses due to "inertia effects". This seems very vague to me. Insofar as it is a key point in justifying this study. The authors should take the time to explain these phenomena in detail?

The effects of inertia on the energy harvesting process can be quite complex and was only very recently investigated systematically by V.D. van Deursen in his MSc project on "Dynamic Simulation Techniques for Airborne Wind Energy Systems", MSc Thesis, TU Delft, September 2024. https://resolver.tudelft.nl/uuid:bb32fc5b-300a-4789-9b48-90927f035378. We have added a link to the MSc thesis and discussed the effects of inertia in more detail.

---

## Author Response (AR2)

**Authors' response to associate editor's comments**

Rishikesh Joshi

September 18, 2024

We appreciate your feedback and comments on our manuscript. Following is our responses.

The provided comments are in the standard font colour black and our responses to the comments are in blue. The associated changes can be seen in the revised manuscript which is submitted with this document.

**Provided minor comments**

1. In my opinion, the authors have addressed the reviewer's comments and I have emailed the authors to point out one final comment, which was a typographical error on line 386 of the revised manuscript.
   Thank you for your agreement with our changes and for identifying the typographical error. Yes, it should be 'wing' and not 'wind'. We have now corrected this.